



# Interannual variations in the $\Delta(^{17}O)$ signature of atmospheric $CO_2$ at two mid-latitude sites suggest a close link to stratosphere-troposphere exchange

Pharahilda M. Steur[1], Hubertus A. Scheeren[1], Gerbrand Koren[2], Getachew A. Adnew[3*], Wouter Peters[1,4], and Harro A. J. Meijer[1]

[1]Centre for Isotope Research (CIO), University of Groningen, Groningen, the Netherlands
[2]Copernicus Institute of Sustainable Development, Utrecht University, Utrecht, the Netherlands
[3]Institute for Marine and Atmospheric research Utrecht (IMAU), Utrecht University, the Netherlands
[4]Environmental Sciences Group, Dept of Meteorology and Air Quality, Wageningen University and Research, Wageningen, the Netherlands
[*]now at the Department of Geosciences and Natural Resource Management, University of Copenhagen, Copenhagen, Denmark

**Correspondence:** Pharahilda M. Steur (p.m.steur@rug.nl)

**Abstract.** $\Delta(^{17}O)$ measurements of atmospheric $CO_2$ have the potential to be a tracer for gross primary production and stratosphere-troposphere mixing. A positive $\Delta(^{17}O)$ originates from intrusions of stratospheric $CO_2$, whereas values close to zero result from equilibration of $CO_2$ and water, predominantly happening inside plants. The stratospheric source of $CO_2$ carrying high $\Delta(^{17}O)$ is, however, not well defined in the current models. More and long-time atmospheric measurements are needed to improve this. We present records of the $\Delta(^{17}O)$ of atmospheric $CO_2$ conducted with laser absorption spectroscopy, from Lutjewad in the Netherlands (53° 24'N, 6° 21'E) and Mace Head in Ireland (53° 20' N, 9° 54' W), covering the period 2017-2022. The records are compared with a 3-D model simulation, and we study potential model improvements. Both records show significant interannual variability, of up to 0.3 ‰. The total range covered by smoothed monthly averages from the Lutjewad record is -0.065 to 0.046 ‰, which is significantly higher than the range of -0.009 and 0.036 ‰ of the model simulation. The 100 hPa 60-90° North monthly mean temperature anomaly was used as a proxy to scale stratospheric downwelling in the model. This strongly improves the correlation coefficient of the simulated and observed year-to-year $\Delta(^{17}O)$ variations over the period 2019-2021, from 0.37 to 0.81. As the $\Delta(^{17}O)$ of atmospheric $CO_2$ seems to be dominated by stratospheric influx, its use a as a tracer for stratosphere-troposphere exchange should be further investigated.

## 1 Introduction

Stable isotope measurements of atmospheric $CO_2$ have been a great asset to carbon cycle research (Mook and Hoek, 1983; Keeling et al., 1984; Ciais et al., 1997; Welp et al., 2011). As different isotopologues of $CO_2$ have the same chemical properties and will be incorporated in the same carbon cycle fluxes, their difference in mass can result in preferred uptake or emission of the lighter or heavier isotopologues for certain processes. This is known as kinetic fractionation (Young et al., 2002). Another form of fractionation is equilibrium fractionation, in which isotopes in different substances at chemical equilibrium





are partially separated (Young et al., 2002). Fractionation will thus influence the isotope composition of atmospheric $CO_2$, and together with $CO_2$ amount fraction measurements, the isotope composition of atmospheric $CO_2$ can help to disentangle carbon sources and sinks (Peters et al., 2018; Welp et al., 2011; Hofmann et al., 2017; Keeling et al., 2005; Laskar et al., 2016). Isotope composition is generally expressed relative to an internationally recognised reference material using the delta notation, according to equation 1:

$$\delta^* A = \frac{[^*A_\mathrm{s}]/[A_\mathrm{s}]}{[^*A_\mathrm{r}]/[A_\mathrm{r}]} - 1 \tag{1}$$

In which A is the atom (for $CO_2$ this is C or O), the superscript * stands for the rare isotope (13 for C, 17 or 18 for O), A without * stands for the most abundant isotope (12 for C, 16 for O) and the subscripts s and r stand for sample and reference, respectively. Delta values are usually expressed in per mille, indicated by the ‰ symbol, as the natural variation is very small. Applications of $\delta(^{13}C)$ and $\delta(^{18}O)$ measurements of atmospheric $CO_2$ are numerous and have proven to be of great value for

identification and quantification of sources and sinks of atmospheric $CO_2$ (Roeloffzen et al., 1991; Ciais et al., 1995; Rayner et al., 2008) and for the description of air mixing dynamics of the troposphere and stratosphere (Assonov et al., 2010).

The relation between $\Delta(^{17}O)$ and $\delta(^{18}O)$, resulting from the kinetic and equilibrium fractionation processes as described above is relatively constant and can be described by:

$$\ln(\delta(^{17}O) + 1) = \lambda \ln(\delta(^{18}O) + 1) \tag{2}$$

With $\lambda$ ranging between 0.5 and 0.53, depending on the dominant fractionation process studied (Adnew et al., 2022). From the $\delta(^{18}O)$ and $\Delta(^{17}O)$ values, or triple oxygen isotope composition, the $\Delta(^{17}O)$, an expression of the deviation from $\lambda$ can be calculated by:

$$\Delta(^{17}O) = \ln(\delta(^{17}O) + 1) - \lambda \ln(\delta(^{18}O) + 1) \tag{3}$$

The $\Delta(^{17}O)$ of tropospheric $CO_2$ is mainly influenced by two processes being 1) intrusion of stratospheric $CO_2$ carrying a

strongly deviating ($\Delta(^{17}O) >> 0$) signal (Thiemens et al., 1995; Boering et al., 2004; Kawagucci et al., 2008; Lämmerzahl et al., 2002), due to the exchange of $CO_2$ and $O_3$ via $O(^1D)$ (Yung et al.), and 2) the equilibration of tropospheric $CO_2$ with water, resulting in $CO_2$ with an $\Delta(^{17}O)$ being close to 0 (Hoag et al., 2005). In this study we use a $\lambda$ of 0.5229 to calculate the $\Delta(^{17}O)$, after the triple oxygen isotope composition of $CO_2$ equilibrated with water by Barkan and Luz (2012). $\Delta(^{17}O)$ is in this case a measure for the deviation from the triple oxygen isotope composition of $CO_2$ equilibrated with water. This equilibration

mainly occurs in plant leaves due to the presence of the enzyme carbonic anhydrase which speeds up the equilibration process of $CO_2$ and water considerably, such that the oxygen isotope composition of $CO_2$ which diffuses from the leaves back into the atmosphere (about 2/3 of the total uptake of $CO_2$ by plants (Adnew et al., 2023)) is largely in equilibrium with that of the leaf water (Francey and Tans, 1987).

Measurements of stable isotopes are traditionally done with isotope ratio mass spectrometry (IRMS), however measurement

of the $\Delta(^{17}O)$ of $CO_2$ is not straightforward with this method due to isobaric interferences of the $^{13}C^{16}O_2$ and the $^{12}C^{16}O^{17}O$ isotopologues. These measurements can therefore only be done by measuring ion fragments, requiring a higher mass resolution





and a very high sensitivity IRMS system, or by $O_2$-$CO_2$ exchange, a sample preparation procedure that is very labor intensive (Adnew et al., 2019). The last method mentioned is at this moment acquiring the highest measurement precision being better than 10 per meg for reference gas measurements of $\Delta(^{17}O)$ (Adnew et al., 2019). Another method for the measurement of

$\Delta(^{17}O)$, next to $\delta(^{13}C)$ and $\delta(^{18}O)$, of atmospheric $CO_2$, directly on atmospheric samples, is laser absorption spectroscopy (Steur et al., 2021). This technique uses the absorption peaks of three different isotopologues of $CO_2$ to define the triple oxygen isotope composition. Therefore, the measurements can be conducted directly on air mixtures containing $CO_2$ at atmospheric amount fractions. This strongly reduces the preparation time for $\Delta(^{17}O)$ measurements, bringing up the potential to set-up large(r) scale measurement programs to evaluate the potential of $\Delta(^{17}O)$ of atmospheric $CO_2$ for carbon cycle and atmospheric

research. From 2017, the Stable Isotopes of $CO_2$ Absorption Spectrometer (SICAS), measuring the $\delta(^{13}C)$, $\delta(^{18}O)$ and $\Delta(^{17}O)$ of atmospheric $CO_2$, has been taken into use at the Centre for Isotope Research (CIO) in Groningen. Air samples from two atmospheric measurement stations, Lutjewad and Mace Head, located at the north coast of the Netherlands and west coast of Ireland, respectively, have been measured regularly at the CIO for their trace gas concentrations and stable isotope composition over the period 2017-2022. We elaborately checked the quality of the measurements by considering the full uncertainty budget,

as well comparing atmospheric sample measurements with results derived from IRMS measurements.

In this paper multi-year records of $\Delta(^{17}O)$ measurements conducted using laser absorption spectroscopy are presented along with the $CO_2$ amount fraction and $\delta(^{13}C)$ and $\delta(^{18}O)$ measurements. Observational data on the triple oxygen isotope composition of troposheric $CO_2$ have been scarcely reported in the literature so far. Earlier records of $\Delta(^{17}O)$ measurements of atmospheric $CO_2$, all conducted using IRMS, have been published before from Jerusalem (Israel) (Barkan and Luz, 2012), La

Jolla (USA) (Thiemens et al., 2014), Taipei (Taiwan) (Liang and Mahata, 2015), cruises on the South China Sea (Liang et al., 2017) and Göttingen (Germany) (Hofmann et al., 2017); the only near-by-mid-latitude measurement site. Göttingen, located about 400 km to the south-west from the Lutjewad atmospheric measurement station in central Germany, has a similar but more continental climate and its record is therefore best comparable to Lutjewad when continental air masses are sampled.

The $\Delta(^{17}O)$ record of Lutjewad has been compared to model simulations of the $\Delta(^{17}O)$ signal of atmospheric $CO_2$ in

Lutjewad as described in Koren et al. (2019). Finally, an outlook is given on how the SICAS, or laser absorption spectroscopy in general, can be used to collect data relevant for studying the $\Delta(^{17}O)$ of atmospheric $CO_2$ in the future.

## 2    Methods

### 2.1    Sampling sites

The Lutjewad atmospheric measurement station is located at the northern coast of the Netherlands, at 53° 24'N, 6° 21'E. Since

2018, Lutjewad station has been a class 2 station in the European Integrated Carbon Observation System (ICOS) network. The station is located directly behind the Wadden Sea dike, in a flat, rural area. The location allows sampling of maritime (background) air with northern winds and continental air (50 % of the time) with southerly winds. Air is pumped from the top of the 60 m high tower via inlets connected to a series of tubing towards a laboratory building containing the instruments for continuous monitoring and an automated flask sampling system. The flasks used in our flask-sampling network are 2.3 L



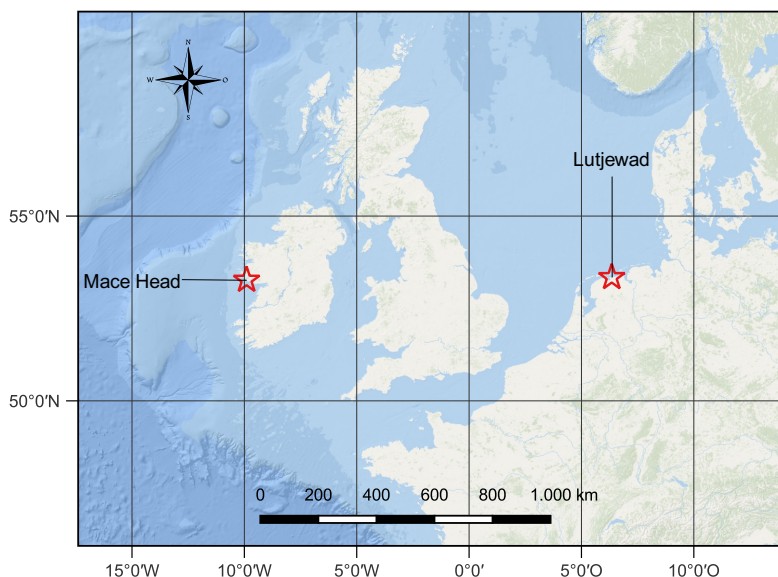

**Figure 1.** Locations on the map of the atmospheric measurement stations Lutjewad and Mace Head (map created by T. Maalderink)

volume glass flasks with two Louwers Hapert Viton sealed valves. The automated flask sampler is able to fill up to 20 flasks at ambient pressure and is set at a typical frequency of 1 flask sample every 3 days taken at 12:00 local time. A flask is flushed for one hour with cryogenically dried sample air to a dewpoint below -50 ° C before the sampler closes it and continuous to flush the next flask (Neubert et al., 2004). Samples of the period 2017-2022 were used for this study. During this period occasionally the flask sampling system failed, causing periods of sparser sampling, especially during 2019 and the beginning of 2022.

The atmospheric measurement station Mace Head, (operated by the National University of Ireland, Galway) is located at the west coast of Ireland (53° 20' N, 9° 54' W) on a cliff at 17 m above sea level. When the wind direction is from the west to southwest, well-mixed air masses from the North-Atlantic cross the station (Stanley et al., 2018). These wind conditions occur about half of the time, and during these periods Mace Head can be used as a background station for Northern Hemisphere background air. Once a week, when the air masses at the site are representative for Northern Hemisphere background air, a flask

sample is taken at the Mace Head station from a 23 m high tower and sent to the CIO for analysis of trace gas measurements and stable isotope composition of $CO_2$. From the beginning of 2019 onwards we started to routinely measure the Mace Head flask samples on the SICAS.

## 2.2 Trace gas amount fraction measurements

Continuous measurements of $CO_2$, $CH_4$, and CO amount fractions have been conducted at the Lutjewad station with cavity

ring down spectrometry (CRDS) (Picarro, G2401 series) since 2013. Flask samples were measured for the same species at the



CIO laboratory, of which the majority was conducted using a customised HP Agilent HP6890N gas chromatograph (HPGC) equipped with a methaniser and a Flame Ionization Detector (Worthy et al., 2003; Laan et al., 2009) in operation until mid-2021, after which we used CRDS for flask analyses. All $CO_2$ measurements are calibrated using in-house made whole dry air working standards linked to the World Meteorological Organisation X2019 scale, $CH_4$ measurements to the X2004A scale, and CO measurements to the X2014A scale. CRDS continuous measurements are shown as hourly means and therefore the standard deviations can vary considerably, depending on the stability of trace gas amount fractions during the measurement period. Flask measurements on the HPGC show typical measurement precisions of <0.1 µmol/mol, <1.0 nmol/mol, and <1.0 nmol/mol for $CO_2$, $CH_4$, and CO, respectively. In addition, CRDS measurements of the flask samples show a typical precision of <0.1 µmol/mol, <0.7 nmol/mol and <2.0 nmol/mol for $CO_2$, $CH_4$, and CO, respectively. The scale uncertainty is ±0.07 µmol/mol for $CO_2$, ±1 nmol/mol for $CH_4$, and ±2 nmol/mol for CO.

### 2.3 Stable isotope measurements

Stable isotope composition measurements are conducted directly on atmospheric air samples with the SICAS, a dual-laser spectrometer (CW-IC-TILDAS-D, Aerodyne) operating in the mid-infrared region. The measurement procedure is extensively described in Steur et al. (2021), so we only briefly explain it here. The calibration procedure and determination of the combined uncertainty differ from the description in Steur et al. (2021) and are therefore more elaborately discussed in this section. A combined uncertainty is determined for all SICAS measurements, including the measurement uncertainty, the repeatability and residual of the measurement series and the introduced uncertainty as a result of the calibration procedure. All these components are explained below.

Measurements are performed in static mode and are repeated for nine aliquots per sample. The gas consumption per aliquot is 20 mL, so measuring one sample requires 180 mL of air. A drift correction is carried out by measuring the working gas (a reference, being a high pressure cylinder containing air of known $CO_2$ amount fraction and $CO_2$ stable isotope composition) continuously, alternating with every aliquot measurement. The standard error of the drift corrected aliquot measurements per sample is the measurement uncertainty.

Besides the flask samples and the working gas we include at least three other references in a measurement series (measure-ment cycle). The references are all measured four times throughout the measurement series. Two of these references, together with the working gas, are used for the calibration of the measurements. One of the references serves as a quality control (QC) measurement, or a known unknown and is not used to determine the calibration curves and is used as an indicator for the quality of the measurement series. The repeatability of the measurement series is calculated as the standard deviation of the four QC measurements. The residual per measurement series is calculated as the average of the calibrated QC measurements minus the known value of the QC.

The calibration method used for a sample measurement depends on the $CO_2$ amount fraction of the sample relative to the references. Analyses of a large number of reference measurements over the years 2020-2022 show that uncertainty introduced by the calibration is highly dependent on the difference, in $CO_2$ amount fraction, of a sample from the closest reference, as well as the difference between the references (for an extensive description and the analyses Chapter 5 in Steur (2023)).We use



two different calibration methods, being the isotopologue method (IM) and ratio method (RM) (Steur et al., 2021), and varying introduced uncertainties are assigned to sample measurements, depending on the difference in $CO_2$ amount fraction between the sample and the references. The IM is used when the sample is within the $CO_2$ amount fraction range of the references. For the IM quadratic calibration curves from the measured isotopologue amount fractions and known amount fractions of a minimum of three references, including the working gas, are determined. Calibrated isotopologue amount fractions of the

samples are subsequently used for the calculation of the delta values. Ideally, the sample is bracketed closely in $CO_2$ amount fraction by the references. When the difference between the nearest reference is 15 µmol/mol or lower, uncertainties of 0.03 ‰ for $\delta(^{13}C)$ and $\delta(^{18}O)$ and 0.05 ‰ for $\Delta(^{17}O)$ and $\Delta(^{17}O)$ are introduced. When the difference is higher than 15 µmol/mol, an uncertainty of 0.09 ‰ for $\delta(^{13}C)$ and $\delta(^{18}O)$ is introduced, and uncertainties of 0.11 and 0.10 ‰ are introduced for $\Delta(^{17}O)$ and $\Delta(^{17}O)$), respectively.

When the sample falls outside of the $CO_2$ amount fraction range of the references, the RM is used. A linear correction of the measured delta values, depending on the $CO_2$ amount fraction is applied. In this way a correction for the introduced $CO_2$ amount fraction dependency of the measured delta values is applied. This correction is needed as a result of measured and assigned isotopologue amount fraction dependencies with a non-zero intercept (Griffith et al., 2012). The uncertainty increases with extrapolation distance, being the difference between the sample and the nearest reference in $CO_2$ amount fraction, with

the sample falling outside the $CO_2$ amount fraction range of the references. The introduced uncertainty (u) in ‰ due to the extrapolation distance ($\Delta y(CO_2)$) in µmol/mol is determined according to the following equations:

$$u_{\delta(^{13}C)} = 0.0042 \Delta y(CO_2) + 0.03\ ‰ \tag{4}$$

$$u_{\delta(^{18}O)} = 0.0054 \Delta y(CO_2) + 0.03\ ‰ \tag{5}$$

$$u_{\delta(^{17}O)} = 0.0063 \Delta y(CO_2) + 0.05\ ‰ \tag{6}$$

$$u_{\Delta(^{17}O)} = 0.0042 \Delta y(CO_2) + 0.05\ ‰ \tag{7}$$

The introduced uncertainties are all based on the empirical data of reference measurements over a period of two to three years, as described in Chapter 5 of Steur (2023). As the Lutjewad and Mace Head stable isotope records presented in this study are measured only on the SICAS, scale uncertainties are not included in the combined uncertainties of the measurements. To prevent showing irrelevant results, only values with a combined uncertainty lower than 0.1 ‰ will be included in the results of

the stable isotope measurements. The highest reference included in the calibration for the records has a $CO_2$ amount fraction of 424.54 µmol/mol, so the consequence is that samples with $CO_2$ amount fractions higher than 441.2, 437.5 and 436.4 µmol/mol are excluded from the $\delta(^{13}C)$, $\delta(^{18}O)$ and $\Delta(^{17}O)$ records, respectively. Especially at the Lutjewad station it is not uncommon to sample air in this range of $CO_2$ amount fractions during winter. This will hence lead to a bias, as results of local or regional events leading to elevated $CO_2$ amount fractions that were captured in the flask records are not included in the results.

For calibration of the SICAS isotope measurements we use the in-house produced gas references, consisting of dried atmospheric air in high pressure gas cylinders, as presented in table 1. The $CO_2$ amount fractions of the references were measured on a PICARRO G2401 gas amount fraction analyzer and calibrated using in-house working standards, linked to the WMO



**Table 1.** Natural air references used for the calibration of stable isotope measurements presented in this study. $\delta(^{13}C)$ and $\delta(^{18}O)$ values as measured at the BGC-IsoLab. $R_{IMAU}$ is the $\Delta(^{17}O)$-$\delta(^{18}O)$ relation as measured at IMAU. $\Delta(^{17}O)$ and $\Delta(^{17}O)$ are calculated from the $\delta(^{18}O)$ and $R_{IMAU}$. Values indicated with * are derived from SICAS measurements as these references were not measured at the IMAU. Uncertainties include the measurement precision and measurement accuracy.

| | $CO_2$ (µmol/mol) | $\delta(^{13}C)$ (VPDB) | $\delta(^{18}O)$ (VSMOW) |
|---|---|---|---|
| Reference 1 | 405.78 ±0.03 | -8.63± 0.013 | 37.269±0.018 |
| Reference 2 | 417.11±0.02 | -9.13±0.019 | 38.102±0.018 |
| Reference 3 | 424.54±0.007 | -9.438± 0.016 | 37.69±0.03 |
| Reference 4 | 413.4 | -8.99±0.012 | 36.884±0.019 |
| Reference 5 | 342.81 | -9.4±0.007 | 37.689±0.017 |
| Reference 6 | 399.08 | -8.22±0.02 | 37.595±0.04 |
| Reference 7 | 378.84±0.05 | -7.52±0.013 | 40.05±0.02 |
| | $R_{IMAU}$ | $\delta(^{17}O)$ (VSMOW) | $\Delta(^{17}O)$ |
| Reference 1 | 0.5218 | 19.280±0.019 | -0.038±0.019 |
| Reference 2 | 0.5215 | 19.693±0.018 | -0.052±0.018 |
| Reference 3 | 0.5216 | 19.49±0.03 | -0.05±0.03 |
| Reference 4 | n.a. | 19.08*±0.06 | -0.04*±0.05 |
| Reference 5 | 0.5211 | 19.464±0.018 | -0.068±0.018 |
| Reference 6 | n.a. | 19.44*±0.06 | -0.04*±0.05 |
| Reference 7 | n.a. | 20.71*±0.15 | -0.04*±0.10 |

2007 scale for $CO_2$ with a suite of four primary standards provided by the Earth System Research Laboratory of the National Oceanic and Atmosphere Administration (NOAA).

170     To ensure cylinders drifting in $CO_2$ amount fraction are identified, all reference cylinders are measured on the PICARRO once a year. The uncertainty of the $CO_2$ amount fractions in table 1 is the standard error of those measurements through the years. Considering the low standard errors ranging between 0.00-0.05 µmol/mol there are no signs of drifting $CO_2$ amount fractions in any of the cylinders.

    Aliquots of all references have been analyzed at the stable isotope lab of the Max Planck- Institut for Biogeochemistry

175 in Jena (BGC-IsoLab) by DI-IRMS to link the $\delta(^{13}C)$ and $\delta(^{18}O)$ directly to the JRAS-06 scale (Jena Reference Air Set for isotope measurements of $CO_2$ in air (VPDB(-$CO_2$) scale)) (Wendeberg et al., 2013). Stable isotope composition of the reference gases measured at the BGC-IsoLab and the standard errors of the measurements (standard error of the results of all aliquot measurements) are presented in table 1.

    Despite the existence of this direct linkage of atmospheric $CO_2$ to the VPDB-$CO_2$ scale, triple oxygen isotope measurements

180 of atmospheric $CO_2$ are usually expressed on the VSMOW scale (Hofmann et al., 2017; Adnew et al., 2020; Boering et al., 2004). Also, an internationally recognized isotope scale for $\Delta(^{17}O)$ has so far not been established. It is therefore not straight-



forward to determine the $\Delta(^{17}O)$ values of our reference cylinders ourselves. We therefore use the $\delta(^{18}O)$ values measured at BGC-IsoLab in combination with triple oxygen isotope measurements of references 1-3 and 5 conducted at the Institute for Marine and Atmospheric research Utrecht (IMAU), using the $O_2$-$CO_2$ exchange method and DI-IRMS measurements (Adnew et al., 2019). The measured $\Delta(^{17}O)$-$\delta(^{18}O)$ relation, calculated as $\ln(\Delta(^{17}O)+1)/\ln(\delta(^{18}O)+1)$ and from now on defined as $R_{IMAU}$, was used to calculate $\Delta(^{17}O)$ values on the VSMOW scale from the $\delta(^{18}O)$ values as measured by BGC-IsoLab. The latter were converted from VPDB-$CO_2$ to VSMOW by the following equation, as recommended in Hillaire-Marcel et al. (2021):

$$\delta_{VSMOW}(^{18}O) = 1.04149 \cdot \delta_{VPDB\text{-}CO_2}(^{18}O) + 41.49 \,‰ \tag{8}$$

Next the following equation was applied to the BGC-IsoLab $\delta(^{18}O)_{VSMOW}$ values:

$$\delta_{VSMOW}(^{17}O) = (\delta_{VSMOW}(^{18}O) + 1)^{R_{IMAU}} - 1 \tag{9}$$

$\Delta(^{17}O)$ values were calculated by applying the $\delta(^{18}O)_{VSMOW}$ and $\Delta(^{17}O)_{VSMOW}$ values to equation 3. For the references that were not measured at the IMAU, the $\delta(^{17}O)$ values were determined from SICAS measurements, using the calibration methods and uncertainty assignment as described before. The $\Delta(^{17}O)$ was subsequently calculated using this measured $\delta(^{17}O)$, and the $\delta(^{18}O)$ as measured by the BGC-IsoLab. Note that the scale described above for the $\Delta(^{17}O)$ values is indirectly linked to VSMOW, adding uncertainty to the compatibility of other $\Delta(^{17}O)$ scales.

For measurement of our reference gases by the BGC-IsoLab and IMAU, aliquots were prepared by connecting 5 sample flasks in series and flush them with the sample gas, resulting in a similar air sample in all flasks. However, deviations of the sampled air and the air in reference cylinders due to alteration of the gas inside the flasks can be introduced (Steur et al., 2023).

## 2.4 Comparison CIO and IMAU

For a selection of Lutjewad samples two flasks containing identical air were sampled (from now defined as a duplo) of which one flask was measured at the CIO and one at the IMAU, to check the compatibility of the $\Delta(^{17}O)$ measurements using laser absorption spectroscopy instead of DI-IRMS. From 2019 a fully automatic extraction system has been taken into use at the IMAU which enables to extract $CO_2$ from air and directly analyse the sample on their DI-IRMS. Before that time, extraction of $CO_2$ from the air samples was done at the CIO and the pure $CO_2$ samples were sent to the IMAU in flame sealed tubes for DI-IRMS analysis.

The $\delta(^{13}C)$ sample differences are higher than expected from the combined uncertainty of the SICAS measurements, as can be seen in figure 2. The frequency distribution shows that at least part of the differences are because of systematic errors, possibly scaling or sampling issues. It should be noted that the quality of the SICAS measurements is lower for the samples measured at the end of 2019 and in 2020. Samples from 2018, from which the $CO_2$ was extracted at the CIO, show a positive offset of the SICAS measurements relative to the IMAU measurements. A reason for the building up of differences in $\delta(^{13}C)$ values during that period has not been found.

Results of the differences of the $\delta(^{18}O)$ measurements are shown in Appendix A1. The differences are far outside the uncertainty range of the SICAS measurements, being up to 2 ‰. These high differences are connected to the observations





of drift in the oxygen isotopes of $CO_2$ in flask samples as a function of time (Steur et al., 2023). $\Delta(^{17}O)$ values are not (or hardly) affected by the drifts in oxygen isotopes in the flasks. We calculated that, in the extreme case of a change of more than 3 ‰ in $\delta(^{18}O)$ of atmospheric $CO_2$ (Steur, 2023), as the result of equilibration of $CO_2$ with water in the flask, changes the $\Delta(^{17}O)$ less than 0.05 ‰. Considering that the uncertainty of the SICAS $\Delta(^{17}O)$ measurements is always 0.05 ‰ or higher, we can conclude that the effect of drift of the oxygen isotopes inside the flasks is negligible for the $\Delta(^{17}O)$ values. Results and

calculations that support this conclusion can be found in Appendix B1.

The $\Delta(^{17}O)$ differences fall in general within the mean combined uncertainty of the SICAS measurements over the whole sampling period. The total range of $\Delta(^{17}O)$ of the samples is, however, small, being 0.15 ‰. Nevertheless, this comparison shows that the CIO calibration procedure gives $\Delta(^{17}O)$ values similar to IMAU, and the repeatability of the measurements falls within the combined uncertainty of the SICAS measurements. The differences are normally distributed so there are no systematic offsets between the two labs.

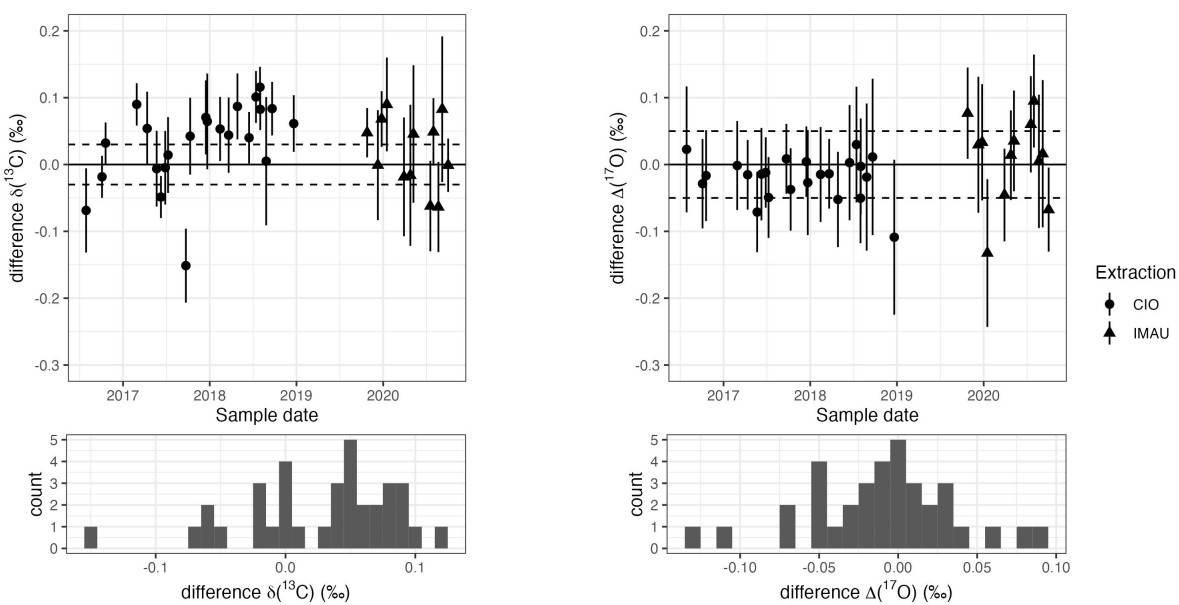

**Figure 2.** The top panels shows the differences (CIO-IMAU) of $\delta(^{13}C)$ and $\Delta(^{17}O)$ measurements of the duplo flasks. Uncertainty bars show the combined uncertainty of the CIO measurements. Shape of the data points indicates whether $CO_2$ was extracted at CIO and send to IMAU as pure $CO_2$ samples (circles) or whether extraction was done at IMAU (triangles). The lower panels shows the frequency distribution of the differences.


In Appendix C we elaborate further on the comparison of $\Delta(^{17}O)$ measurements and also the complete record of Lutjewad $\Delta(^{17}O)$ measurements, including all IMAU measurements. This figure shows that the variation that is observed in the IMAU measurements coincides with variation that is observed from the SICAS measurements.



## 2.5 Atmospheric modeling of $\Delta(^{17}O)$ in $CO_2$ at Lutjewad

In addition to the measurements, we present model simulations for $\Delta(^{17}O)$ in $CO_2$ for the Lutjewad location by the 3-D
transport model described in Koren et al. (2019). As the Mace-Head and Lutjewad latitude is similar, we do not expect to see
significant differences of simulations between the two location. The model includes the stratospheric input of high $\Delta(^{17}O)$
and processes that will lead to reduction of the stratospheric signal, being biosphere activity, equilibration of $CO_2$ with soil
moisture, $CO_2$ emissions from fossil fuel and biomass burning and $CO_2$ uptake and emission from the oceans. An update to this
model is the use of meteorological driver data from the ERA5 release (Hersbach et al., 2020), instead of the ERA-Interim fields
(Dee et al., 2011) that were used previously. The model resolution applied for the Lutjewad simulation is a longitude-latitude
grid of 1° by 1°. Here we present simulations for $\Delta(^{17}O)$ in $CO_2$ for the years 2017 until the beginning of 2021. Note that
the long-term mean values simulated by the model for Lutjewad are ultimately dependent on the integrated contribution from
all processes across the globe, which are poorly constrained in the model (e.g. due to large uncertainties in soil exchange, see
Wingate et al. (2009)). Therefore we focus on the timing and amplitude of the seasonal cycle and the interannual variability of
$\Delta(^{17}O)$ in $CO_2$ at the Lutjewad station.

## 3 Results and discussion

### 3.1 $CO_2$ amount fraction measurements results

In figure 3 the $CO_2$ amount fraction measurements of the Lutjewad flasks over the period 2017-2022 are shown, together with
the continuous $CO_2$ amount fraction measurements from Lutjewad. In addition the $CO_2$ amount fraction measurements of
the Mace Head flasks measured at the CIO are shown. As an independent comparison, the Mace Head $CO_2$ amount fraction
measurements of discrete air samples from the NOAA-Global Monitoring Laboratory Carbon Cycle Greenhouse Gases coop-
erative air sampling network (NOAA-GML CCCSN) (Lan et al., 2022) are also plotted in the same figure. The Mace Head
flask samples measured at the CIO are in good agreement (within precision) with the overlapping time series results from
the NOAA-GML CCGG. A background $CO_2$ amount fraction curve has been determined from the Lutjewad continuous $CO_2$
amount fraction measurements. This was done by including only measurements of samples taken during daytime (between
10:00 and 19:00 UTC) and excluding hourly averages with standard deviations higher than 0.5 μmol/mol and CO values higher
than 140 nmol/mol. Subsequently, the filtered signal was smoothed by a moving average of 30 points and the result was fitted
with a quadratic trend with a 2-harmonic seasonality. The resulting background signal, shown in figure 3 as the black line, cor-
responds well with the Mace Head measurements. This confirms that the derived background curve represents well-mixed air,
not influenced by local contamination events from fossil fuel burning. This $CO_2$ amount fraction background curve is used in
the stable isotope records to calculate the $\Delta_{bg}y(CO_2)$ of a sample, being the difference in amount fraction between the sample
and the background curve.

  The background curve shows the strong influence of the biosphere, resulting in $CO_2$ amount fractions that are almost 15
μmol/mol lower in summer than in winter. The seasonality shows maxima at the beginning of the growing season in March and





April and minima at the end of the growing season in August. The overall increase of $CO_2$ amount fractions in the atmosphere is clearly visible from the background curve and is 2.5 µmol/mol per year. These results are very close to growth rate of the globally averaged $CO_2$ amount fractions of 2.4 µmol/mol (standard deviation 0.5 µmol/mol per year) per year from 2011 to 2019 (Friedlingstein et al., 2022).

The Lutjewad flasks, although sampled at noon with the aim to sample well-mixed tropospheric air, occasionally show large positive deviations from the background curve, especially in winter, of up to +47 µmol/mol in December 2017. The $CO_2$ enriched signals are most probably due to local and regional sources that use fossil fuels carried to the Lutjewad station from the continent. We therefore expect to see more deviations from the seasonal cycles of stable isotope values induced by the more continental influence at the Lutjewad record when compared to the Mace Head record.

The Europe wide drought, which was most severe in Northern Europe, during the summer of 2018 (Peters et al., 2020; Ramonet et al., 2020) is clearly visible in the continuous $CO_2$ amount fraction record of Lutjewad, as a deviation from the overall decrease in amount fractions that normally occurs over the growing season. In early spring of 2018, $CO_2$ amount fractions decrease rapidly (when the growing conditions were more favorable, see Smith et al., (Smith et al., 2020)), until May 2018, when a rapid increase in $CO_2$ amount fractions is observed that lasts until June, before $CO_2$ amount fractions start

decreasing again. This event is only visible in one Lutjewad flask sample having a $\Delta_{bg}y(CO_2)$ of -8.6 µmol/mol. Due to the too low sampling frequency, the drought event is hard to identify from the flask samples only. In 2022 Europe experienced another severe drought, which was, however, mostly located in central and southeastern Europe (van der Woude et al., 2023). This drought event does not show up in the continuous amount fraction record of Lutjewad as we see for the 2018 drought.

### 3.2    Stable isotope measurements

Results of $\delta(^{13}C)$ measurements on atmospheric $CO_2$ of discrete flask samples from Lutjewad and Mace Head are shown as a function of sampling date in figure 4. A quadratic trend with a 2-harmonic seasonality is fitted on all Lutjewad $\delta(^{13}C)$ points that have a $\Delta_{bg}y(CO_2)$ that is not higher than 5 µmol/mol. There are too few data points for doing a fit on the Mace Head record. Instead, the Lutjewad seasonal trend is also plotted in the Mace Head record so both records can be easily compared. The seasonality in the $\delta(^{13}C)$ records shows a strong anti-correlation with the $CO_2$ amount fraction records, with maxima during

late summer, and minima in late winter. During winter, there are negative excursions in the Lutjewad record that do not appear in the Mace Head record, from which we can conclude that the Lutjewad $\delta(^{13}C)$ is influenced by more local or regional signals resulting in more depleted $\delta(^{13}C)$ signals like fossil fuel emission and plant and soil respiration (Keeling et al., 2017; Scholze et al., 2008). For the same reason, the seasonal curve derived from the Lutjewad data shows a stronger decrease of $\delta(^{13}C)$ values in Autumn and Winter than the Mace Head record. When Lutjewad would be more influenced by the strong biosphere

activity at the continent, heavier (i.e. less negative) $\delta(^{13}C)$ values than at Mace Head would be expected during the summer. For the year 2020 and 2021 this is the case, but in 2019 the Mace Head values are heavier. The year 2019 is, however, a period in which Lutjewad samples were collected more sparsely due to problems with the sampling system. It is therefore hard to conclude whether there is in general a heavier $\delta(^{13}C)$ signal in Lutjewad compared to Mace Head during the summers. Overall,





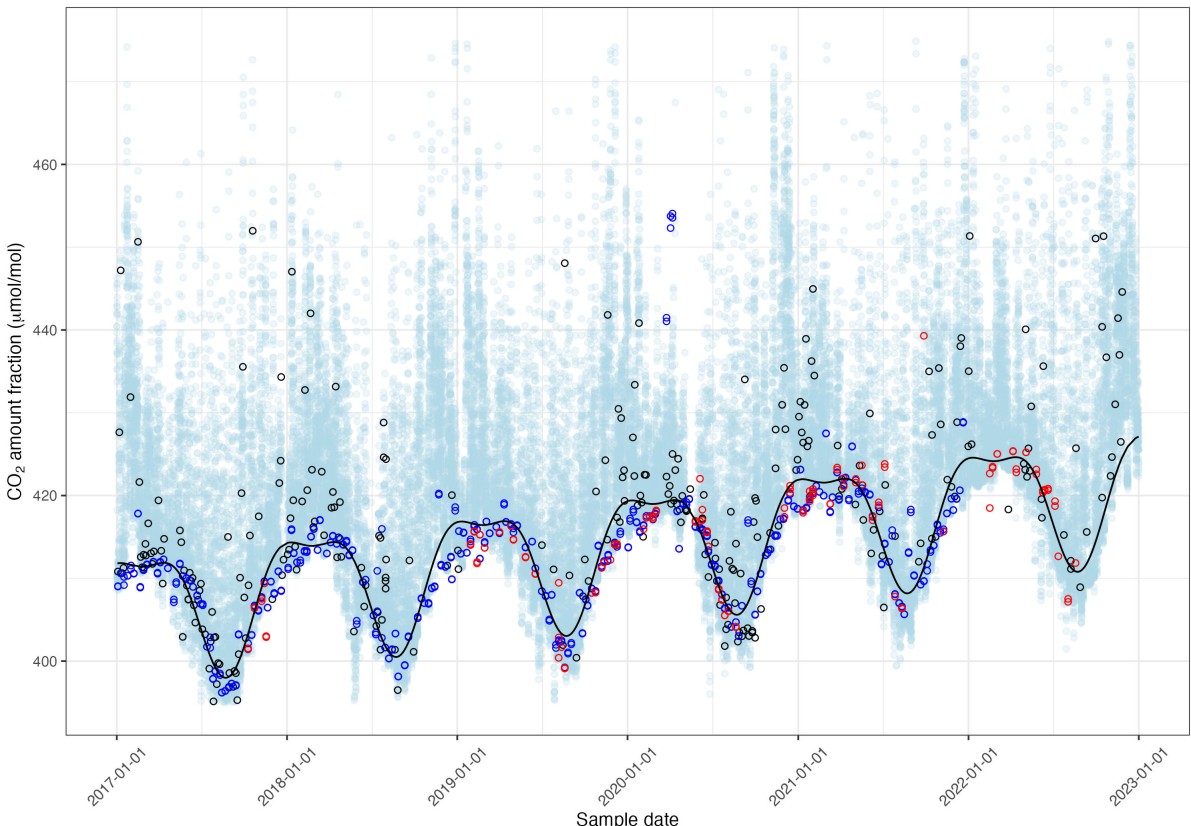

**Figure 3.** CIO $CO_2$ amount fractions from continuous measurements shown as hourly averages (light blue points) and discrete flask sample measurements (black points) of the Lutjewad atmospheric measurement station, and discrete flask sample measurements from the Mace Head atmospheric measurement station from the NOAA GML Carbon Cycle Cooperative Sampling Network (dark blue points) and the CIO (red points). The seasonal cycle (black line) is derived from the filtered continuous measurements that were fitted with a quadratic trend with a 2-harmonic seasonality.

a decreasing trend is observed from the seasonal fit, explained by the increase in $CO_2$ amount fractions in the atmosphere due
to the combustion of fossil fuels, also known as the $^{13}$C Suess effect (Keeling, 1979).

Measurements of $\delta(^{18}O)$ of atmospheric $CO_2$ from Lutjewad and Mace Head flask measurements conducted at the SICAS are presented in figure 5. A seasonal curve is fitted from the Lutjewad data using the same method as for the $\delta(^{13}C)$ data. The Mace Head observations coincide with the Lutjewad fit, with maxima in May and June and minima in December and January. Although the maximum and minimum values are very similar, the maximum values in Mace Head during the summer of 2022
are more enriched than the Lutjewad values. These differences might be explained by the difference in $\delta(^{18}O)$ composition of the source waters for the vegetation at both sites (Levin et al., 2002). It should be noted that a great part of the $\delta(^{18}O)$ values of the atmospheric $CO_2$ samples shown here are likely to have a bias towards depletion, due to the drift we observe over time,





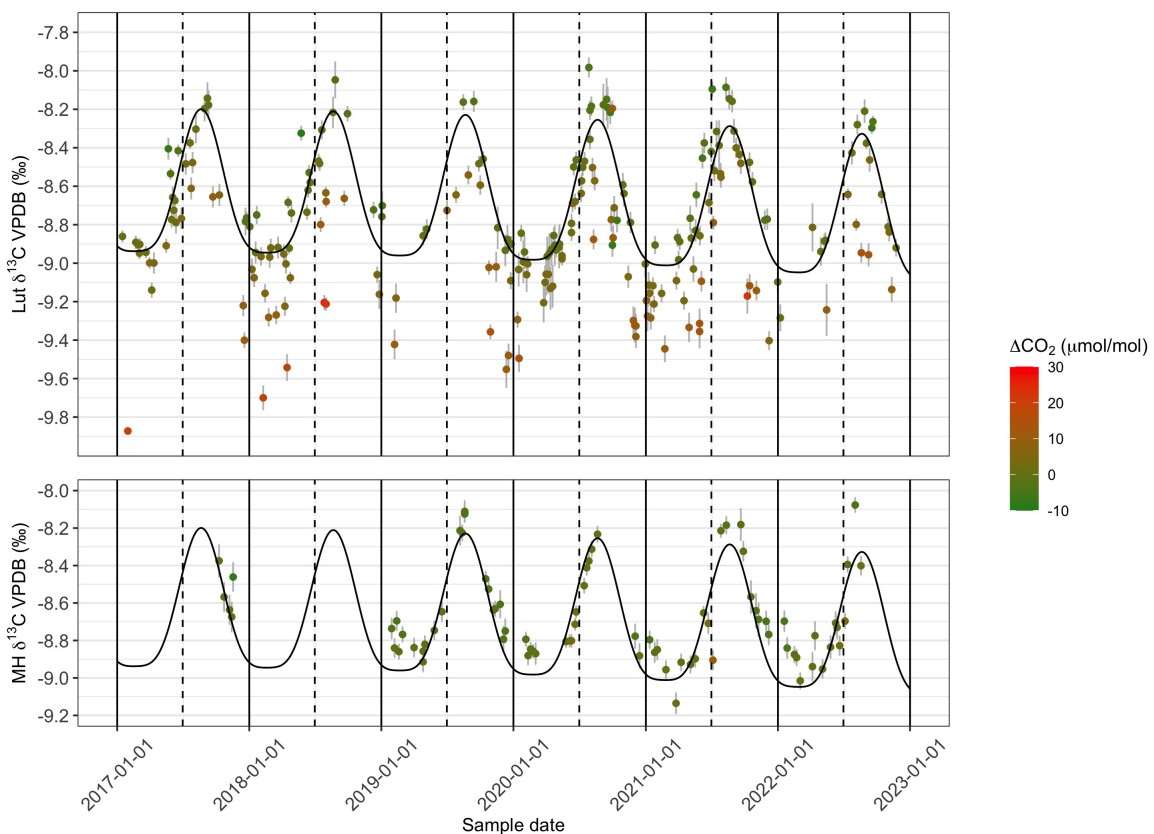

**Figure 4.** $\delta(^{13}C)$ records of Lutjewad and Mace Head from SICAS flask measurements of atmospheric $CO_2$. The upper graph shows $\delta(^{13}C)$ measurements from Lutjewad, the lower graph from Mace Head. The combined uncertainty of the measurements is shown as the grey error bars and include measurement uncertainty, repeatability and accuracy and introduced uncertainty as a consequence of the calibration method used. $\Delta_{bg}y(CO_2)$ is indicated by the colour of the data points, with red being positive deviations and green negative deviations. The seasonality curve is derived from fitting the Lutjewad $\delta(^{13}C)$ values of samples that have a $\Delta_{bg}y(CO_2)$ not higher than 5 μmol/mol and is shown as the black line in both graphs (the Lutjewad seasonal curve is also shown in Mace Head graph). The fitting method that was used is the same as for the $CO_2$ background curve.

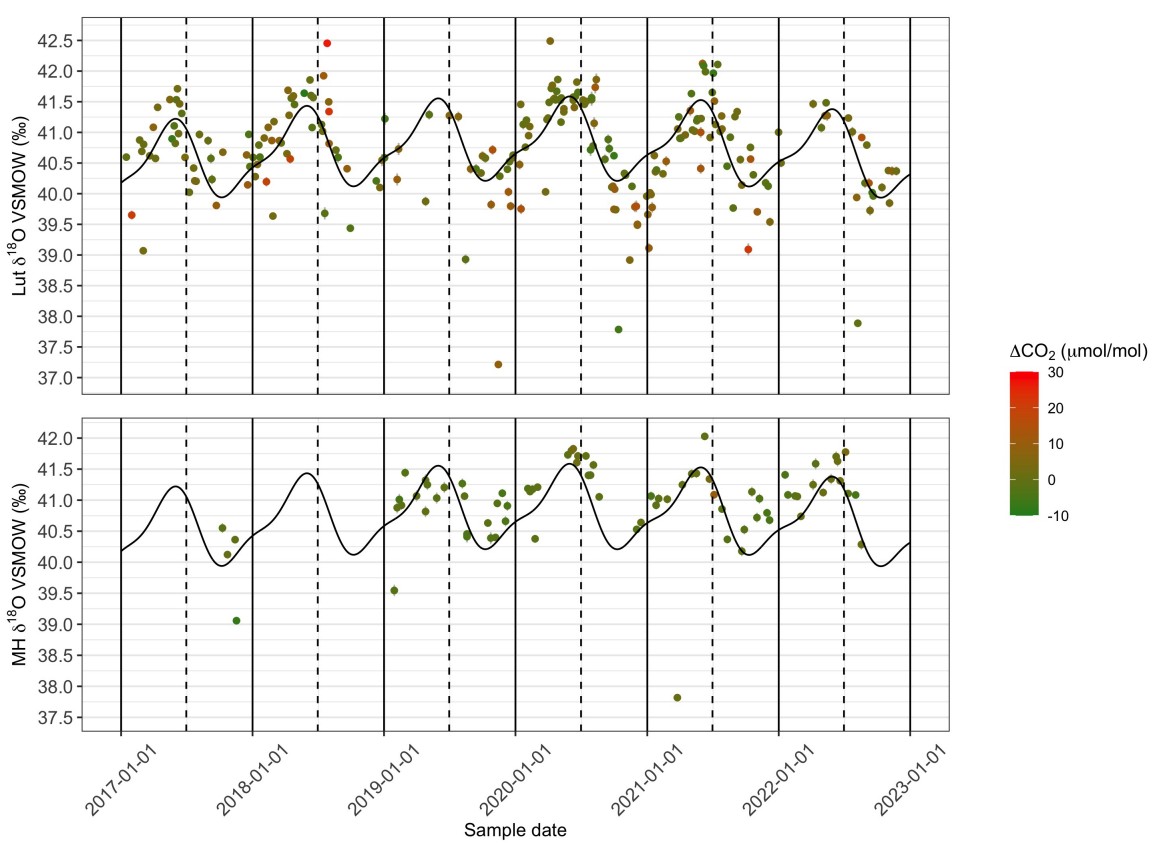

**Figure 5.** $\delta(^{18}O)$ records of Lutjewad (upper graph) and Mace Head (lower graph) from SICAS flask measurements of atmospheric $CO_2$. The combined uncertainties are shown as the grey error bars and include measurement uncertainty, repeatability and accuracy and introduced uncertainty as a consequence of the calibration method used. $\Delta_{bg}y(CO_2)$ is indicated by the colour of the data points, with red being positive deviations and green negative deviations. The seasonality curve is derived from fitting the Lutjewad $\delta(^{18}O)$ values of samples that have a $\Delta_{bg}y(CO_2)$ not higher than 5 µmol/mol and is shown as the black line in both graphs (the Lutjewad seasonal curve is also shown in Mace Head graph). The fitting method that was used is the same as for the $CO_2$ background curve.





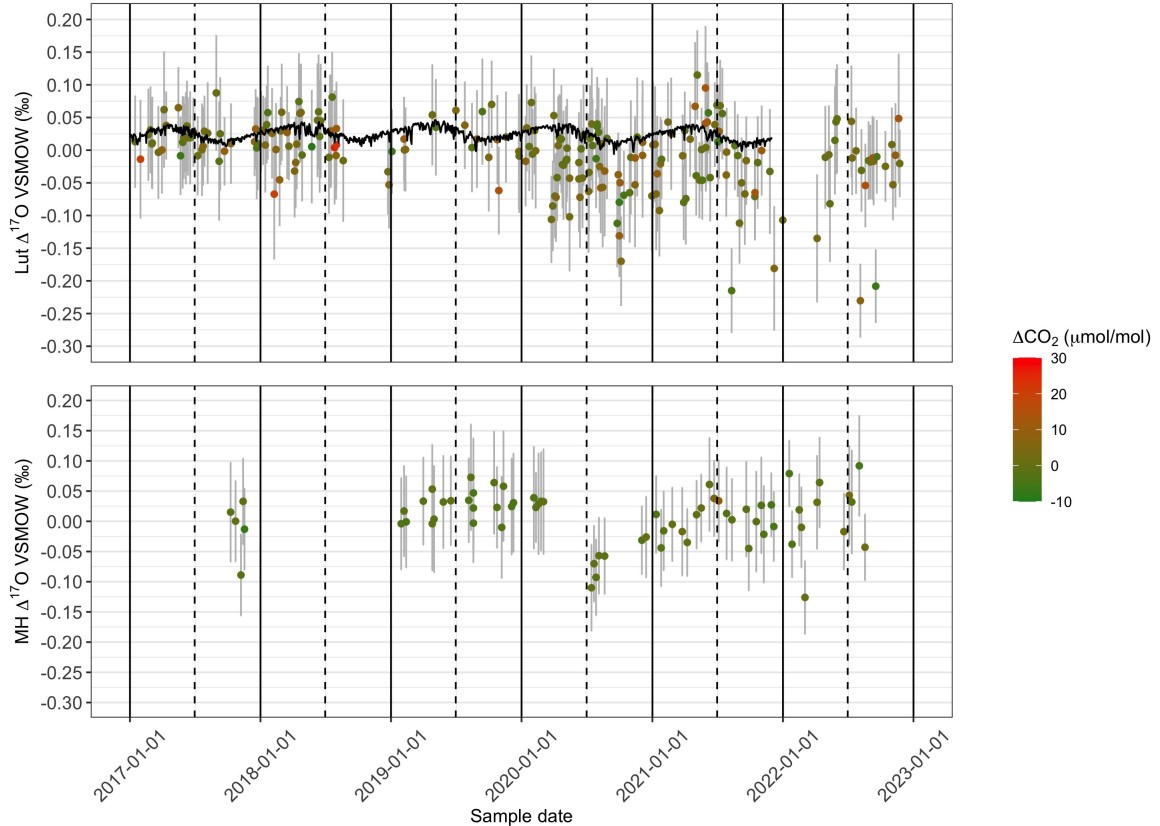

**Figure 6.** $\Delta(^{17}O)$ records of Lutjewad (upper graph) and Mace Head (lower graph) from SICAS flask measurements of atmospheric $CO_2$. The combined uncertainties are shown as the grey error bars and include measurement uncertainty, repeatability and accuracy and introduced uncertainty as a consequence of the calibration method used. $\Delta_{bg}y(CO_2)$ is indicated by the colour of the data points, with red being positive deviations and green negative deviations. Model simulations for Lutjewad (Koren et al., 2019) showing daily values at 13:00 UTC are shown in the upper graph as the black line.

explained in section 2.4. Interpretation of the absolute changes in the $\delta(^{18}O)$ values should therefore be done with caution, taking the sample age into account.

$\Delta(^{17}O)$ measurements from the Lutjewad and Mace Head stations are presented in figure 6. In both records the majority of the values occur within a range of +0.1 to -0.1 ‰. The average combined uncertainty of all measurements is 0.07 ‰, and lies therefore very close to the overall signal that is present in the records. Following Hoag et al. (2005) we would expect to see a clear seasonality of increasing $\Delta(^{17}O)$ values over winter and decreasing values over spring and summer, when the biosphere is active. Such a seasonality is not detected in the Lutjewad and Mace Head $\Delta(^{17}O)$ records. This differs from results from the

Göttingen record over the period 2010-2012, where a seasonality was observed with maximum values during June and July (Hofmann et al., 2017). The amplitude of the seasonality that was determined from the Göttingen $\Delta(^{17}O)$ record is (0.13±0.02)



‰ in agreement with the range of +0.1 to -0.1 ‰ for the majority of our record. If such a seasonality would be present in the Lutjewad and Mace Head record, we would expect to see it, as this signal is higher than the average combined uncertainty of the SICAS measurements. It can be, that due to the more continental location, the amplitude of the $\Delta(^{17}O)$ seasonality is higher at the Göttingen site reflecting a stronger biosphere signal. If a lower seasonality amplitude would be present at Lutjewad and Mace Head, it is not detected so far due to too high measurement uncertainty.

The low sampling frequency in combination with the sampling method at both locations will complicate capturing small seasonal variations. Air samples represent only a snapshot in time, while at the same time the frequency of sampling is only once every 3 days (Lutjewad) or even once every week (Mace Head) (Nevison et al., 2010). Changing the sampling method to a method in which the sampled air evenly represents a certain sampling period, would decrease the influence of short variability in the atmospheric composition at the sampling site (Chen et al., 2012). To get a much higher sampling frequency for $\Delta(^{17}O)$ measurements at Lutjewad in the future, our laser absorption spectroscopy system will be deployed in the (semi-)continuous measurement mode, a technique already shown by Kaiser et al. (2022). This will enable us to apply rigid filtering on the data to derive either results representative of well-mixed background air, or, on the other hand, results that are representative for air masses from the continent.

The most important difference between the two sites is the presence of more depleted values in the Lutjewad record, with the lowest value being -0.23 ‰ in the summer of 2022. In summer, leaf water gets enriched in oxygen isotopes, and depleted in $\Delta(^{17}O)$ as the result of high rates of transpiration (Landais et al., 2006). Due to the active biosphere during summer, $CO_2$ and leaf water will equilibrate and the depleted $\Delta(^{17}O)$ signal will be translated to the $CO_2$ (Adnew et al., 2023). Half of the points that are more negative than $\Delta(^{17}O)= -0.1$ were sampled during (late) summer, and can therefore be explained by this mechanism. In winter, when there is no strong influence of an active biosphere, nor high rates of transpiration, we should seek for other potential processes causing the depleted $\Delta(^{17}O)$ values. $CO_2$ emitted from combustion processes has very negative $\Delta(^{17}O)$ values (Laskar et al., 2016; Horváth et al., 2012). The influence of $CO_2$ enrichment events due to fossil fuel emissions, which are captured regularly during the winter months in the Lutjewad flask record, could therefore explain the more depleted $\Delta(^{17}O)$ values. All points that have lower $\Delta(^{17}O)$ than -0.1 ‰, and are sampled during winter/spring, have more depleted $\delta(^{13}C)$ values and more enriched $CO_2$ values than would be expected from the seasonal trends. This indicates that local $CO_2$ emission sources are the reason for the more depleted $\Delta(^{17}O)$ values in winter. Samples that are very enriched in $CO_2$ amount fractions are not shown here, as that results in very high measurement uncertainties. It is possible that if proper calibration of these measurements would have been possible, an even clearer signal of $CO_2$ derived from combustion processes would be observed.

Significant differences of $\Delta(^{17}O)$ values over time are observed in both records. In the Lutjewad record we observe $\Delta(^{17}O)$ values that are above or close to 0 at the beginning of 2020. Then values decrease until reaching minimum values in October 2020 with the lowest value in that period being -0.17 ‰. An increase of $\Delta(^{17}O)$ values is observed after this period and May 2021 is a period with more elevated values, with the highest observation being +0.12 ‰. Although the Mace Head record shows gaps over the period from 2020-2021, it is clear that values at the beginning of 2020 are higher than at the end of 2020.





This interannual variability in both records indicates that other processes than biosphere activity cause most of the variation at the measurement locations.

### 3.3 Sensitivity analysis simulated $\Delta(^{17}O)$ inter-annual variability

The total variation predicted by a local simulation of the model (base version in Koren et al., 2019) for the location of Lutjewad
is lower than the uncertainty and variability of the SICAS measurements, and has a seasonal character only. The model simulation of $\Delta(^{17}O)$ of atmospheric $CO_2$ is shown for the Lutjewad mid-latitude band as the black line in figure 6. Daily values at 13:00 UTC (corresponding to 14:00 or 15:00 local time) are shown, that thus represent well-mixed afternoon conditions, which are typically more reliable in relatively coarse global models than simulated nighttime or early morning values. Although small, there is a clear seasonality with highest $\Delta(^{17}O)$ values in early spring when stratospheric influx is highest with low biospheric
activity aggregated over the preceding months, and lowest values during early autumn, when the biospheric carbon uptake has depleted the tropospheric $\Delta(^{17}O)$ budget. Values are all between 0.036 and -0.009 ‰ which is significantly narrower than the observed range at Lutjewad. It is therefore clear that the current model version does not capture the variation in $\Delta(^{17}O)$ that is measured at the Lutjewad record. The Göttingen record (Hofmann et al., 2017) also shows significant inter-annual changes in $\Delta(^{17}O)$ values of 0.1 ‰, which have not been explained so far. In that record, spanning the period from June 2010 until August
2012, they found a negative shift in the $\Delta(^{17}O)$ values from the summer of 2011 until the end of the record.

We do not consider the biosphere sink of $\Delta(^{17}O)$ to cause the inter-annual changes in the records since then a stronger or weaker seasonal cycle is also expected to occur. The variations observed in the Lutjewad and Mace Head records is furthermore driven by anomalies in multiple seasons and not limited to summer or winter periods only. Besides the biospheric sink, the other main term in the $\Delta(^{17}O)$ budget is its stratospheric production and downward transport. We therefore hypothesized that
the stratospheric input of $\Delta(^{17}O)$ is not well parameterized in the model, leading to the limited inter-annual variability that is simulated in figure 6.

In the 3-D atmospheric model the stratospheric production of $\Delta(^{17}O)$ of atmospheric $CO_2$ is implemented using its empirical relation with stratospheric $N_2O$ amount fractions (see Sect. 2.2 in Koren et al., 2019, for a more detailed description), shown in the equation below

$$\Delta_{\text{fit}}(^{17}O) = a(y_{\text{dtd}}(N_2O) - 320.84 \text{ nmol/mol}) + b \tag{10}$$

where $\Delta_{\text{fit}}(^{17}O)$ is the assigned stratospheric signature, $[N_2O]_{\text{dtd}}$ is the detrended $N_2O$ amount fraction, and $a$ and $b$ are empirical fit coefficients. The $N_2O$ amount fraction in the stratosphere and the $\Delta(^{17}O)$ in $CO_2$ are negatively correlated, based on measurements from four different studies, as presented in Koren et al. (2019). The use of this relation as the only driver for the $\Delta(^{17}O)$ source from the stratosphere is very coarse and it is possible that factors, such as temperature, as postulated by Wiegel
et al. 2013, have an effect on the $\Delta(^{17}O)$ enrichment of $CO_2$ in the stratosphere.

To increase variability in the simulated stratospheric production of $\Delta(^{17}O)$, we included an additional empirical production term for the region 60-90° N in winter based on 100 hPa temperature anomalies (over this same period and region) from the National Centers for Environmental Prediction (NCEP). Temperature at 100 hPa at 60-90° N, or lower stratosphere temperature,





is shown to be a proxy for stratosphere-troposphere exchange during the months January-March, as it is linked to the strength
of the polar vortex, which negatively correlates with the strength of the Brewer-Dobson circulation (Newman et al., 2001;
Nevison et al., 2007). The Brewer-Dobson circulation is the meridional circulation driven by large-scale temperature gradients
on earth leading to ascending air near the tropics and subsidence of air near the poles. A strong Brewer-Dobson circulation
will lead to higher volumes of stratospheric air intruding into the troposphere during winter, as the polar vortex is then weaker
(Nevison et al., 2007). This links the lower stratosphere temperature during the Northern-Hemisphere winter to the strength of
the input of $CO_2$ with a high $\Delta(^{17}O)$ composition into the troposphere. The adjusted $\Delta(^{17}O)$ production term is defined as

$$\Delta_{\text{source}}(^{17}O) = \overbrace{a(y_{\text{dtd}}(N_2O) - 320.84 \text{ nmol/mol}) + b}^{\text{original production term}} + \overbrace{c\Delta T_{100\text{hPa}}}^{\text{added term}} \tag{11}$$

where the first part is repeated from equation 10, and the added last term describes the imposed coupling with temperature
anomalies at the 100-hPa level $\Delta T_{100\text{hPa}}$ with $c$ being a tunable parameter. Here, the empirical parameters $a$, $b$ and $c$ are constant
for a given simulation, whereas the $[N_2O]_{\text{dtd}}$ value differs for each grid point and with time. The temperature anomaly $\Delta T_{100\text{hPa}}$
in this equation is set to zero for regions below 60° N and for months April-December; and for the months January-March
the 100-hPa temperature anomaly (for 60-90° N) is averaged. Note that the temperature relation represents both temperature
dependence of the actual $\Delta(^{17}O)$ as suggested in Wiegel et al. (2013) and the temperature dependence in stratospheric exchange,
which might not be sufficiently represented with only 25 vertical layers in the current model (see e.g. Bândă et al., 2015, for
the influence of vertical resolution on stratosphere-troposphere exchange).

The simulation with the $\Delta(^{17}O)$ adjusted production term is in much better agreement then the original Lutjewad simulation,
as can be seen in figure 7. It is striking that over the years 2019-2021 the model simulation follows the running average of the
measurements very well with a correlation coefficient of 0.81 for this period. For the years 2017 and 2018 the data and the
model agree in that there is much less inter-annual variability in both the measurements and the model simulation. However
on the small scale variability the comparison is hampered by the relatively high uncertainty of the measurements. The overall
variability over the full record is -0.061 to 0.056 ‰ for the model simulation and -0.065 to 0.046 ‰ for the moving average
of the measurements, so the adjusted $\Delta(^{17}O)$ production term increased the overall variability of the model significantly. The
much improved agreement of the model simulation and the measured $\Delta(^{17}O)$ indicates the need for revising the stratospheric
source of the model, now done using the empirical relation with stratospheric $N_2O$ amount fractions. This will lead to a more
realistic input of high $\Delta(^{17}O\text{-}CO_2)$ into the troposphere.

The added source production term linked to lower stratosphere temperature is chosen for the adjusted model simulation
because of its connection to troposphere-stratosphere transport at higher latitudes, as well as its relation to ozone concentrations
in the stratosphere. A weak polar vortex is accompanied by elevated stratospheric temperatures and more stratosphere to
troposphere downwelling, while low stratosphere temperatures lead to a more stable polar vortex and therefore downward
transport (Newman et al., 2001; Kidston et al., 2015). On top of that, we should consider the role of ozone amount fractions
in the stratosphere on the formation of $CO_2$ with high $\Delta(^{17}O)$ values. Formation of polar stratospheric clouds, accelerating the
destruction of ozone, is known to occur during anomalous cold conditions, when the polar vortex is strong and stable (Lawrence
et al., 2020). We hypothesize that colder lower stratosphere temperatures at 60-90° N lead to enhanced ozone destruction in





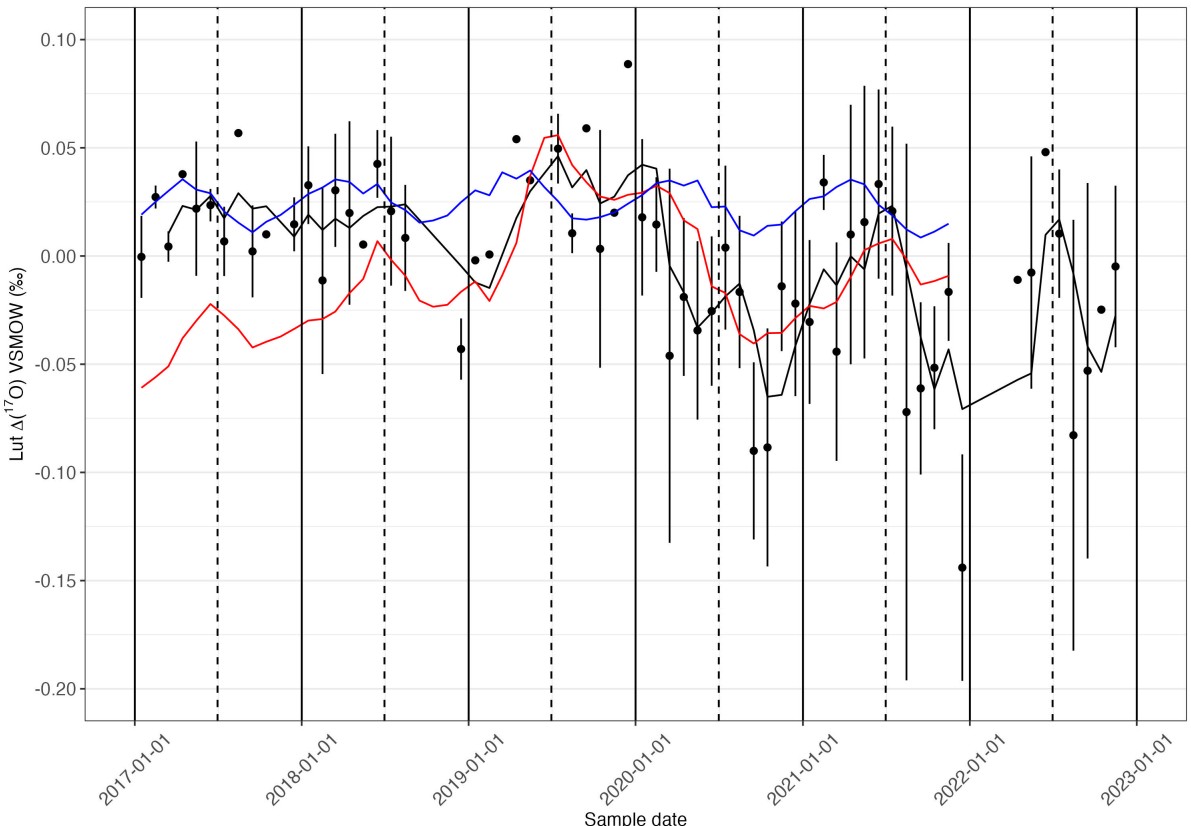

**Figure 7.** In black dots the monthly averages of the Lutjewad $\Delta(^{17}O)$ record, the error bars show the standard deviation of all values per month, plotted with a running average (window=3) in the black line. In blue the monthly averages of the original model simulation (as described in (Koren et al., 2019). In red the monthly averages of the model simulation in which the input of $\Delta(^{17}O)$ into the troposphere is linked to the lower stratosphere temperature. For better visual comparison of the plots, we subtracted 0.08 ‰ from the latter plot.

the polar vortex which in return might reduce the production of high $\Delta(^{17}O\text{-}CO_2)$ in the stratosphere during late winter and early spring in the Northern-Hemisphere (given the role of ozone in the production of $\Delta(^{17}O)$ stratospheric $CO_2$, Yung et al.).

415     This will result in generally lower $\Delta(^{17}O)$ values in tropospheric $CO_2$ after that period. This ozone-dependent process would than add up to the atmospheric transport process to reduce the $\Delta(^{17}O)$ budget in the troposphere. The considerations above indicate that especially in more anomalous stratospheric conditions, both at higher and lower than average temperatures, the stratospheric $\Delta(^{17}O)$ source is likely to deviate from the linear fit of stratospheric $N_2O$ amount fractions and $\Delta(^{17}O)$ values in $CO_2$ as was used in Koren et al. (2019). We acknowledge that our empirical modification still does not accurately describe the

420     intricate complexities of the stratospheric production of $\Delta(^{17}O)$, but at least allows us to assess the relevance of the stratospheric source for the model simulation and points in a direction where further model improvements can be beneficial.



Summarizing, we do observe interannual changes in both the Lutjewad and Mace Head records, possibly caused by variations in the stratospheric source of enriched $\Delta(^{17}O)$. No seasonal cycle, which would be an expected effect of the biosphere, is observed but stratosphere-troposphere exchange seems to cause the highest variations in $\Delta(^{17}O)$.

## 4   Conclusion and outlook

In this study we showed that $\Delta(^{17}O)$ measurements of atmospheric $CO_2$, as well as $\delta(^{13}C)$ and $\delta(^{18}O)$ measurements, can be performed using laser absorption spectroscopy, thereby drastically reducing sample preparation time in comparison with IRMS measurements. This opens the opportunity to do long-term monitoring studies or field studies more easily and can lead to an increase in $\Delta(^{17}O)$ measurements of atmospheric $CO_2$ in the near future. With our analysis method we reach combined uncertainties of 0.05 ‰ when the sample $CO_2$ amount fraction is within the range of the reference gases, and the sample does not differ more than 15 μmol/mol from the nearest reference. Extrapolation of the calibration curves, or high differences between the sample and the nearest reference introduce uncertainty of the results, showing the importance of including enough reference gases in the calibration. For $\delta(^{13}C)$ and $\delta(^{18}O)$ measurements we reach combined uncertainties of 0.03 and 0.05 ‰ respectively, under good calibration conditions. Seasonal cycles, as well as long-term trends as can be expected from the known sources and sinks of $^{13}C$ and $^{18}O$ were clearly identified from the Lutjewad and Mace Head measurement records. The $\Delta(^{17}O)$ records show significant interannual variability at both measurement locations. A seasonal cycle is not observed, possibly due to too high uncertainties of the measurement results. A better precision can be reached by improving the alignment and by adjustment of the spectral fit. The measurement instrument will be used for semi-continuous measurements at the Lutjewad station in the near-future, resulting in much higher sampling frequency so rigorous filtering can be applied on the measurement results. In this way we hope to link variations in the records to specific events which will help us understand the $\Delta(^{17}O)$ budget of atmospheric $CO_2$ in the troposphere better.

As original model simulations do not capture the interannual variability as observed in the measurements, we revised the model definition of stratospheric input of $CO_2$ with a high $\Delta(^{17}O)$ value into the troposphere by linking it to temperature anomalies of the 60-90° N lowermost stratosphere as a proxy for the downwelling strength. This resulted in a much stronger inter-annual variability in the model simulation for the Lutjewad location, following the variations in $\Delta(^{17}O)$ measurements closely for the years 2019-2021. This suggests that the inter-annual variability in the tropospheric budget of $\Delta(^{17}O)$ as observed in the Lutjewad measurements is more strongly coupled to year-to-year variations in stratospheric downwelling of enhanced $\Delta(^{17}O)$ values in $CO_2$ than previously assumed. Our results show that the biosphere is not the dominant process for variations in $\Delta(^{17}O)$. $\Delta(^{17}O)$ of atmospheric $CO_2$ is therefore not suitable as a proxy for quantification for gross primary production at our study locations. The variation in the stratospheric source of high $\Delta(^{17}O)$ is possibly the cause for the high interannual variations we observe in the records. We should therefore consider the potential of using $\Delta(^{17}O)$ of atmospheric $CO_2$ as a proxy for stratospheric intrusion instead.



*Data availability.*   Data presented in this paper can be downloaded from https://doi.org/10.34894/1XJG1F

## Appendix A: $\delta(^{18}O)$ differences SICAS and IMAU flask measurements

$\delta(^{18}O)$ measurements of samples that should be identical conducted at the IMAU and with the SICAS differ strongly, as can be seen in figure A1. We argue that this is due to drift of the oxygen isotopes in the flasks, a phenomenon described in (Steur et al., 2023). In figure A2 the difference in $\delta(^{18}O)$ (SICAS-IMAU) is plotted against the difference in storage time, for the flasks of which the date of $CO_2$ extraction is known. Especially the samples of which the $CO_2$ was extracted at the IMAU show a negative trend, as expected from the fact that over time $\delta(^{18}O)$ values drift towards more negative values. The samples

extracted at the CIO do not show this trend, possibly due to the fact that another extraction method was used, adding more uncertainty to the $\delta(^{18}O)$ values. The time that passed before samples were measured at the IMAU after extraction at the CIO is very variable. It is possible that the pure $CO_2$ samples drifted over time. This extra uncertainty factor is not included in this analysis.

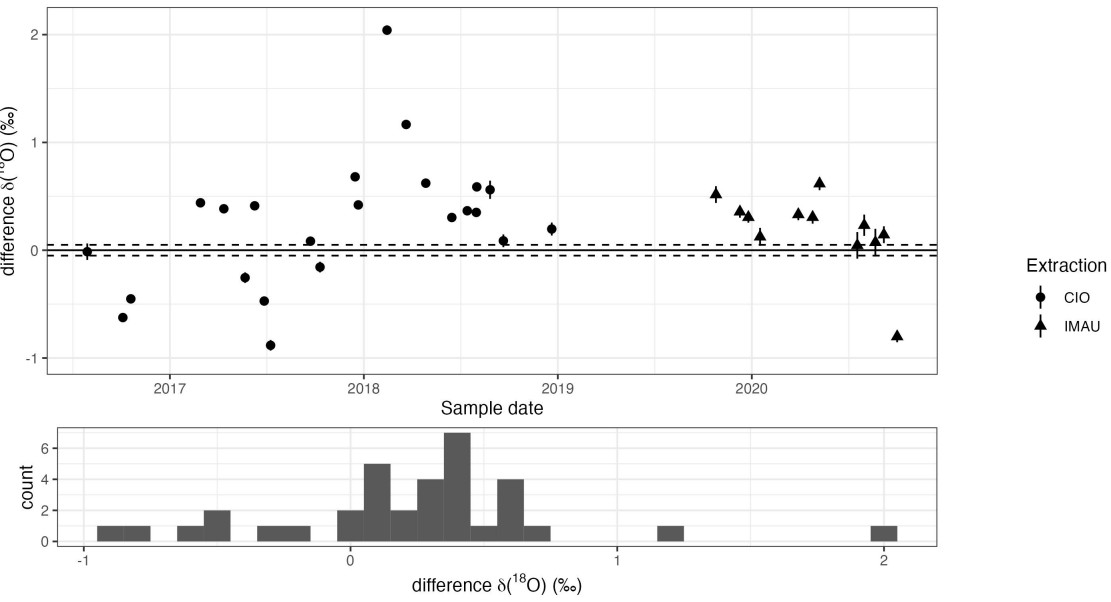

**Figure A1.** The top panel shows the differences (CIO-IMAU) of $\delta(^{18}O)$ measurements of the duplo flasks. Uncertainty bars show the combined uncertainty of the CIO measurements. Shape of the data points indicates whether $CO_2$ was extracted at CIO and send to IMAU as pure $CO_2$ samples (circles) or whether extraction was done at IMAU (triangles). The lower panels shows the frequency distribution of the differences.





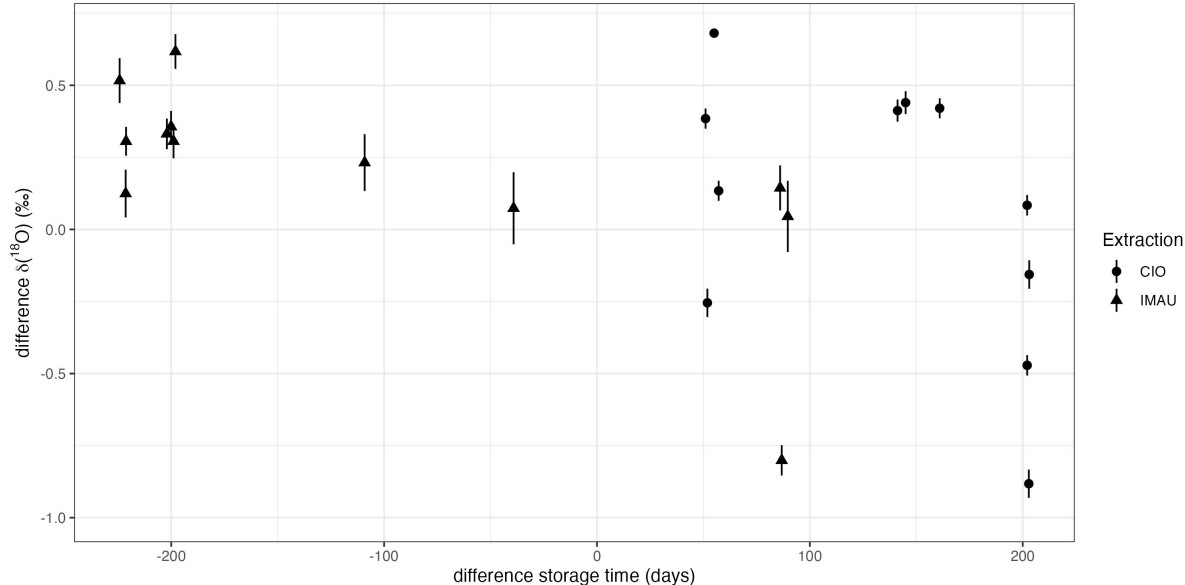

**Figure A2.** The differences of $\delta(^{18}O)$ measurements (CIO-IMAU) of the duplo flasks is plotted against the difference in storage time. Uncertainty bars show the combined uncertainty of the CIO measurements. The shape of the datapoints indicates whether samples were extracted at the CIO (circles) or at the IMAU (traingles).

## Appendix B: Sensitivity analysis drift of $\Delta(^{17}O)$ in glass sample flasks

To determine the change in $\Delta(^{17}O)$ as the result of drift of the oxygen isotopes of atmospheric $CO_2$ inside glass sample flasks (Steur et al., 2023), a simulation of the various changes was conducted. In an earlier study we showed that small amounts of water inside the flasks will change the original oxygen isotope composition of the atmospheric $CO_2$, as $CO_2$ and water will equilibrate. Water builds up inside the flasks over time, both through sampling and through permeation of water into the flask through the Viton O-rings (Steur et al., 2023). For the simulation we use flasks of 2.3 L containing air with a $CO_2$ amount

fraction of 400 μmol/mol at atmospheric pressure. The initial $\delta(^{18}O)$ of the atmospheric $CO_2$ is 37 ‰ on the VSMOW scale, within the atmospheric range. The $\Delta(^{17}O)$ varies such that there are simulations for initial $\Delta(^{17}O)$ values of 0.5, 0.1, 0, -0.1 and -0.5 ‰. For the initial $\delta(^{18}O)$ and $\Delta(^{17}O)$ of the water we use -12.91 and -6.77 ‰ VSMOW, respectively. These values are measurement results of extracted water from lab air. We assume that all $CO_2$ and water equilibrate over time. The amount of water inside the flask varies between $10^{-4}$ and $10^{-6}$ g, such that equilibration causes changes in $\delta(^{18}O)$ of the atmospheric $CO_2$

between -3.27 and -0.03 ‰. It should be noted that changes of more than 3 ‰ in $\delta(^{18}O)$ are very high, as changes of 0.48 ‰ were observed after 114 days of storage in similar conditions as described above (Steur et al., 2023). The change in $\Delta(^{17}O)$ of the atmospheric $CO_2$ ranges between -0.05 and 0.01 ‰ for all scenarios described above (see also figure B1).





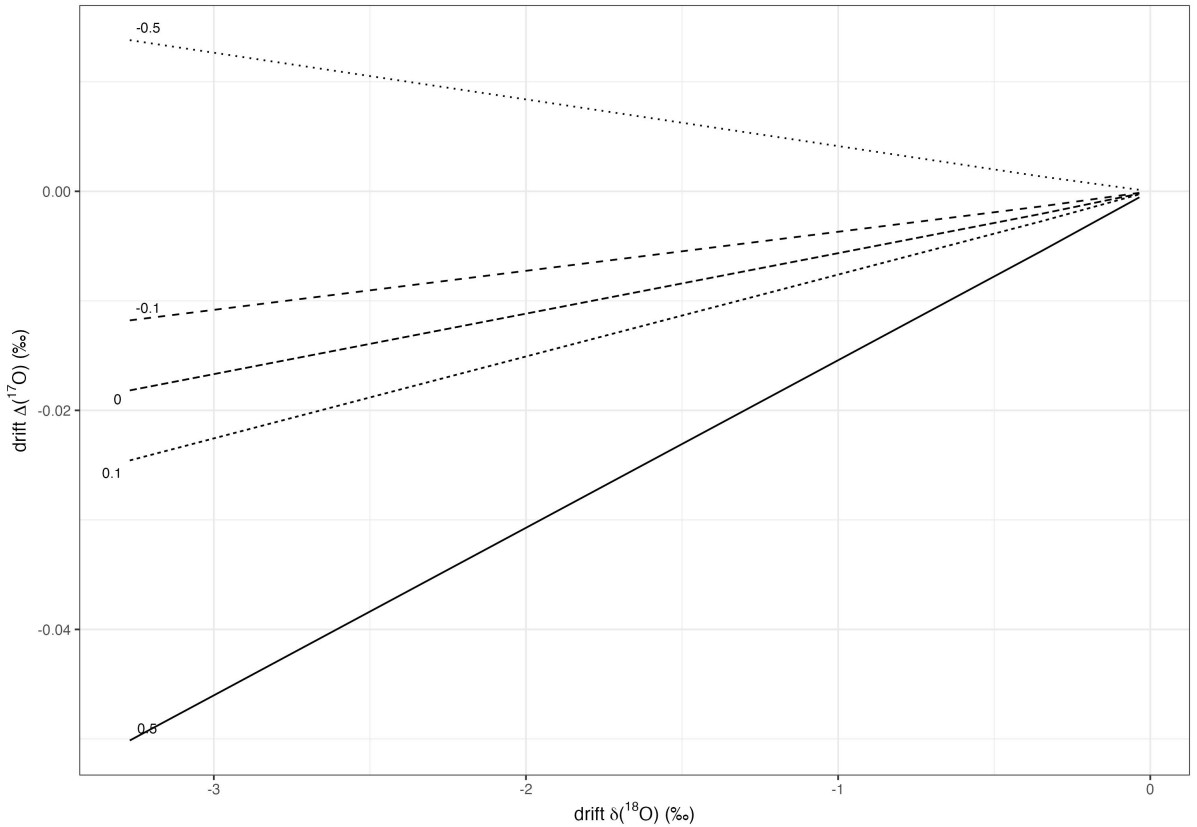

**Figure B1.** Results of a sensitivity analysis of the drift of $\Delta(^{17}O)$ of atmospheric $CO_2$ in a 2.3 L glass sample flask as a function of the drift in $\delta(^{18}O)$. The initial $\Delta(^{17}O)$ value of the atmospheric $CO_2$ is indicated per line.

## Appendix C: Lutjewad $\Delta(^{17}O)$ measurements from CIO and IMAU compared

Two identical flask samples (duplo's) have occasionally been taken at Lutjewad, with the aim to compare measurements of the
SICAS with IRMS measurements from IMAU. For $\Delta(^{17}O)$ measurements this comparison is hard to make, considering the
very low variance that is observed in the duplo samples from the Lutjewad station. Also, not all duplo's have been measured
by both labs. Figure C1 shows results of identical samples measured at IMAU and CIO in a space representative for the total
range in $\Delta(^{17}O)$ that was measured at Lutjewad. From the figure it is clear that the identical samples that were measured are
not representing the full range of $\Delta(^{17}O)$, ranging from -0.2 to 0.1 ‰. We should also consider the (undefined) uncertainty
added by the extraction of the $CO_2$ from air, which is done for the IMAU measurements, but not for the SICAS measurements.

When considering, however, all datapoints of the Lutjewad flasks from the SICAS and from the IMAU, significant interannual variability is reflected in both datasets. Both the IMAU and the SICAS measurements show lower values in 2020 than in the period before, as observed in figure C2.





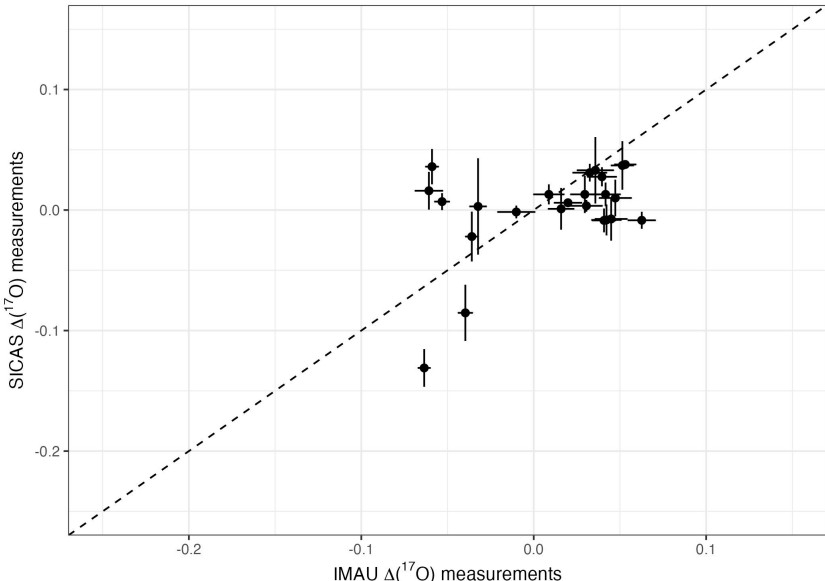

**Figure C1.** $\Delta(^{17}O)$ measurements conducted with IRMS at IMAU (x-axis) and conducted with the SICAS (y-axis), all in ‰. The error bars show the standard errors of the measurements. The dashed line is the 1:1 ratio.



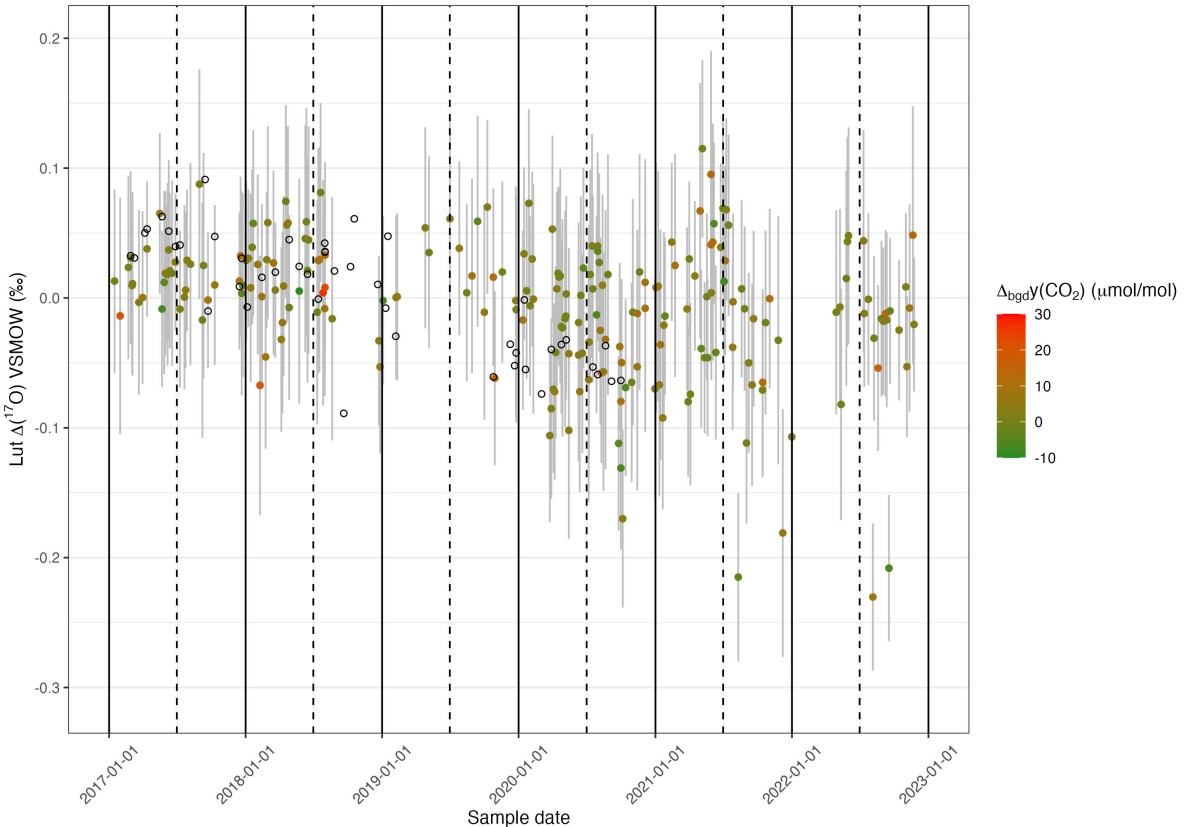

**Figure C2.** $\Delta(^{17}O)$ record of Lutjewad from SICAS flask measurements (filled circles) and DI-IRMS flask measurements of duplo flasks from the IMAU (open circles). The combined uncertainties of the SICAS measurements are shown as the grey error bars and include measurement uncertainty, repeatability and accuracy and introduced uncertainty as a consequence of the calibration method used. The difference in amount fraction between the sample and the background curve, or $\Delta CO_2$, is indicated by the colour of the data points, with red being positive deviations and green negative deviations.

*Author contributions.* PS and HS conducted the spectral measurements. GA conducted the IRMS measurements. PS did the data analysis.
GK performed the model simulations. PS wrote the text, HS and GK gave input for the discussion. All authors helped to finalise the manuscript.

*Competing interests.* The authors declare that they have no conflict of interest.



*Acknowledgements.* We thank Gerard Spain from the University of Galway for sampling of the Mace Head flasks for many years. Stable isotope composition measurements of the SICAS calibration gases were conducted at the Max Plank Institute for Biogeochemistry in Jena, and we thank Heiko Moossen and his team for that. The simulations were performed on the HPC cluster Aether at the University of Bremen, financed by DFG within the scope of the Excellence Initiative. This project has received funding from the EMPIR programme co-financed by the Participating States and from the European Union's Horizon 2020 research and innovation programme. WP, GA, and GK acknowledge funding from the European Research Council, for the ASICA project under grant (649087).



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
