# Peer review of "Interannual variations in the $\Delta$ (17O) signature of atmospheric CO2 at two mid-latitude sites suggest a close link to stratosphere-troposphere exchange"

_EGUsphere, 2023_

## Author Comment (AC1)

Answers to referee #1:

We thank the referee for the recommendations and valuable comments that helped to improve the quality of the paper, as well as the compatibility of the data we present. Below our reactions on the specific comments can be found in the **bold and italic text.**

Specific comments:

Suggest that the authors change "$\lambda$" in Equation 2 to "$\Theta$ ". Equation 2 refers to specific kinetic processes with unique isotope fractionation factors. Such physical variables are commonly designated as theta "$\Theta$" values in the literature to distinguish them from the slope "$\lambda_{RL}$" in Equation 3, which is an arbitrary number.

***We agree with this suggestion and changed the $\lambda$ to $\Theta$ in equation 2, and, for consistency, changed the $R_{IMAU}$ to $\Theta_{IMAU}$ in section 2.3.***

Strongly suggest the authors recalculate and report the $\Delta'^{17}O$ values using a $\lambda_{RL}$ = 0.528 instead of 0.5229. First, water triple oxygen papers use $\lambda_{RL}$ = 0.528, and since the composition of air $CO_2$ is closely linked to water compositions, it is reasonable to use the same $\lambda_{RL}$. Second, the triple oxygen isotope community is now adopting $\lambda_{RL}$ = 0.528 as a consensus value, independent of the field of study and materials analyzed; see Miller & Pack (2021). Using the consensus value of 0.528 will make comparing the presented data with existing and future literature easier.

***We now use $\lambda_{RL}$ = 0.528 in all the reported $\Delta('^{17}O)$ values, as this will indeed make comparing the data with literature easier. It has to be noted that calibration of $CO_2$ and water will now give a $\Delta('^{17}O)$ of -0.21 ‰ instead of 0.***

In line 53, the authors state that $O_2$–$CO_2$ exchange currently provides the highest measurement precision triple O data. It may be worth noting that multiple papers in recent years have demonstrated sub-10 ppm precision for $CO_2$ measurements using laser spectroscopy (e.g., Bajnai et al., 2023; Hare et al., 2022; Perdue et al., 2022; Stoltmann et al., 2017).

***We thank the reviewer for the suggested references. We added these techniques and references and changed the text to (line 52): "These measurements can therefore only be done by measuring ion fragments, requiring a higher mass resolution and a very high sensitivity IRMS system, or by $O_2$-$CO_2$ exchange, a sample preparation procedure that is very labor intensive (Adnew et al., 2019; Mahata et al., 2013). The last method mentioned is at this moment acquiring a precision higher than 10 per meg for measurements of $\Delta'(^{17}O)$ (Adnew et al., 2019; Liang et al., 2023). Laser absorption spectroscopy measurements of $\Delta'(^{17}O)$ (next to $\delta(^{13}C)$ and $\delta(^{18}O)$) on pure $CO_2$ (Stoltmann et al., 2017) and directly on $CO_2$-in-air (Steur et al., 2021; Hare et al., 2022; Perdue et al.,***

**2022; Bajnai et al., 2023) now reach precisions close to, or higher than the IRMS measurements. "**

To the paragraph starting with line 130: It may be worth noting that Perdue *et al.* (2022) doesn't observe a shift in $\Delta'^{17}O$ values related to the $pCO_2$ mismatch between the sample and reference (see their Fig. 8), whereas Bajnai *et al.* (2023) does, but they correct it by precisely matching the $pCO_2$ of the reference to the sample (see Fig. 4). It seems that the mismatch in $pCO_2$ between the sample and reference is the largest source of uncertainty in the presented data. As an outlook, could the authors discuss how to make their measurements more precise?

**We have attempted to provide the requested outlook, and new text was added to elaborate on the influence of the $CO_2$ amount fraction on the measurement uncertainty:**

**Line 138 "The calibration method used for a sample measurement depends on the $CO_2$ amount fraction of the sample relative to the references. The uncertainty introduced by the calibration is highly dependent on the difference, in $CO_2$ amount fraction, of a sample from the closest reference, as well as the difference between the references (Steur, 2023). We calibrate with the reference cylinders only, instead of having an on-line mixing facility where the reference and sample $CO_2$ amount fraction can be matched (Perdue et al., 2022; Bajnai et al., 2023). Therefore, samples that fall outside the range of the $CO_2$ amount fraction that is covered by our reference cylinders will have higher uncertainties."**

**In line 171 we added the sentence: "Extending the $CO_2$ amount fraction range of our reference cylinders will improve the measurement precision of samples with elevated $CO_2$ amount fractions, as well as extend the range of $CO_2$ amount fractions that can be shown in the results. A way to prevent that a high number of reference cylinders has to be included at all times, is to make the selection of references more dynamic. As sample measurements are always alternated with a working gas measurement, it is possible to do a 1-point calibration immediately after a sample is measured. In this way it will be possible to select the ideal set of references to calibrate the samples based on the $CO_2$ amount fractions derived from the 1-point calibration. This would save reference gas, as well as measurement time of a measurement series."**

Bajnai *et al.* (2023) observed the dependence of the TILDAS-$\Delta'^{17}O$ data on the measurement temperature. While the reference bracketing method used in the presented dataset likely addressed such temperature variations, the authors could further increase the credibility of their data by discussing this effect in the paper.

**The reference bracketing method should indeed correct for instrumental drift caused by temperature variations. Although the temperature effect specifically has not been studied for the SICAS, we do provide analysis of stability of the measurements over a period of 2 years in Steur (2023). The standard deviations found in this analysis are included in the combined uncertainty of all our measurements. We therefore do not separately address the temperature effect in this manuscript.**

In lines 267 and 336, the authors argue that $CO_2$-enriched signals are due to the contribution of fossil fuel emissions. In this case, one would expect to see correlations between $\delta^{13}C$, $\Delta'^{17}O$, and $pCO_2$. Can the authors underline their statements with data and possibly additional figures?

*Based on this request, we have evaluated the available data in more detail and decided to change our text to weaken the statement. The $\delta^{13}C$ and $CO_2$ amount fraction logically correlate which is also clear from figure 4, but a Keeling plot (not shown) did not distinguish natural from anthropogenic sources in this region, as signatures are too similar. We added a new figure in the Appendix showing the $\Delta'(^{17}O)$ summer and winter values of Lutjewad plotted against 1/$CO_2$ and $\delta^{13}C$. We see no correlation in these plots, and we therefore make no conclusions on the exact $CO_2$ sources in the Lutjewad record. The statements were adjusted in line 278:*

*"The Lutjewad flasks, although sampled at noon with the aim to sample well-mixed tropospheric air, occasionally show large positive deviations from the background curve, especially in winter, of up to +47 µmol/mol in December 2017. The CO2 enriched signals are most probably due to local and regional sources of CO2, either natural or anthropogenic, that occur on the continent. We therefore expect to see more deviations from the seasonal cycles of stable isotope values induced by the more continental influence at the Lutjewad record when compared to the Mace Head record."*

*And in line 350: "The most important difference between the Lutjewad and Mace Head $\Delta'(^{17}O)$ records is the presence of more depleted values in the Lutjewad record, with the lowest value being -0.43 ‰ in the summer of 2022. CO2 equilibrated with water following $\lambda_{RL}$ will have an $\Delta'(^{17}O)$ of -0.21 ‰. In summer, leaf water gets enriched in oxygen isotopes, and depleted in $\Delta'(^{17}O)$ as the result of high rates of evapotranspiration (Landais et al., 2006). Due to the active biosphere during summer, CO2 and leaf water will equilibrate and the depleted $\Delta'(^{17}O)$ signal will be translated to the CO2 (Adnew et al., 2023). We estimated that this could result in $\Delta'(^{17}O)$ values being up to 0.1 ‰ more depleted, when assuming the minimum θ of 0.516 for evapotranspiration (Landais et al., 2006), and considering the range of $\delta(^{18}O)$ values that were measured in our Lutjewad record. For the full estimation we refer to Appendix E. $\Delta'(^{17}O)$ values up to -0.31 ‰ can be explained by this process. CO2 emitted from combustion processes has very negative $\Delta'(^{17}O)$ values (Laskar et al., 2016; Horváth et al., 2012). All points that have lower $\Delta'(^{17}O)$ than -0.3 ‰, and are sampled during winter/spring, have more depleted $\delta(^{13}C)$ values and more enriched CO2 values than would be expected from the seasonal trends. This indicates that local CO2 emission sources are the reason for the more depleted $\Delta'(^{17}O)$ values in winter. Samples that are very enriched in CO2 amount fractions are not shown here, as that results in very high measurement uncertainties. This could be the reason that a correlation of $\Delta'(^{17}O)$ and CO2 amount fractions does not appear in figure D1. A few points show depletions lower than -0.31 ‰ without CO2 amount fraction enrichments, and remain for now unexplained."*

In the paragraph starting with line 310, the authors argue that they should be able to resolve a 130 ppm annual variation in $\Delta'^{17}O$, as observed by Hoffman *et al.* (2017). However, their argument that their uncertainty of ±100 ppm is lower than 130 ppm is misleading and needs to be revised. Instead, the authors should take into account the signal-to-noise ratio and the number of measurements to determine what cyclic signal can be resolved in their time series.

***The uncertainty of our measurements is (on average) 70 per meg. The 100 per meg is the range in which the majority of the points fall. We agree with the reviewer that this is a confusing statement, and we changed it to the following:***

***Line 320: "$\Delta'(^{17}O)$ measurements from the Lutjewad and Mace Head stations are presented in figure 6. The total range in the Lutjewad and the Mace Head record is 0.5 and 0.2 ‰, respectively, with an average combined uncertainty of the measurements of 0.07 ‰ for both records."***

In line 315, the authors write that the amplitude of the seasonal $\Delta'^{17}O$ signal in Göttingen is larger due to a stronger biosphere signal. This is an important statement in comparing the presented record with existing data and thus should be expanded upon. Would the 3-D model used in this paper be able to reproduce the 130 ppm signal observed by Hoffman *et al.* (2017)?

***We thank the reviewer for this suggestion. We now compared the amplitude of the model simulation from Hoffman et al. (2017) for the Göttingen location and the model simulation of Lutjewad, conducted with the model described in Koren et al. (2019). The results were added as the following text:***

***Line 330: "The amplitude of the seasonality that was determined from the Göttingen $\Delta'(^{17}O)$ record is (0.13±0.02) ‰. If such a seasonality would be present in the Lutjewad and Mace Head record, we would expect to see it, as this signal is higher than the average combined uncertainty of the SICAS measurements. It can be, that due to the more continental location, the amplitude of the $\Delta'(^{17}O)$ seasonality is higher at the Göttingen site reflecting a stronger biosphere signal. A model simulation of the Göttingen location shows an amplitude of 0.045 ‰ (Hofmann et al., 2017), while the amplitude of the simulation at the Lutjewad location, shown as the black line in figure 6 is close to 0.025 ‰. The model used in the Hofmann paper (2017) is an earlier version of the model used in this study (Koren et al., 2019), so the results should be well comparable. The higher amplitude for the simulation of the Göttingen location confirms the hypothesis of a higher $\Delta'(^{17}O)$ seasonality due to the more continental location in comparison with Lutjewad. It is unlikely that a lower seasonal signal than observed at the Göttingen location would be detected by the SICAS measurements considering their average combined uncertainties"***

The following changes are suggested for Figures 4, 5, and 6: The range of the top and bottom plots should be the same, which will help the reader make visual comparisons easily. The measurement locations should be written above the curves

and not on the vertical axis label. The coloring of the $\Delta p\text{CO}_2$ should be changed to a diverging, color-blind-friendly color scale.

***We thank the reviewer for these suggestions that we all applied to the plots.***

Please note the following suggestions for improving the visibility of Figure 7: The vertical year-markers should be made thinner so that they don't clash with the data and error bars. The red trend should be plotted accurately without any shift by 0.08‰ to avoid confusion. Moreover, the horizontal axis grids, similar to those in Excel-made figures, are unnecessary for any plots.

***The changes suggested here were also applied to the plot, and the colors of the lines in all plots were changed to a color blind friendly palette.***

Suggest adding Carlstad & Boering (2023) to the list of references in line 15.

***The reference was added to the list.***

The sentence in line 437, "A better precision...", is without precedence in the text. The authors may consider either expanding on it or removing it.

***This sentence was removed.***

Correct the spelling in line 87: "continues".

***The spelling was corrected.***

References cited in this

---

## Author Comment (AC2)

Reaction to referee #2:

We thank the referee for his/her comments that helped improving the manuscript. We agree with his/her general comment that SICAS measurement results can and should be improved for future research. This is something that we have been working on in the last year and we expect to gather measurements of higher quality in the near future. Despite the low $\Delta'(^{17}O)$ variability in the atmosphere and the relatively high uncertainty of our measurements published here, the interannual changes in the $\Delta'(^{17}O)$ records of Lutjewad and Mace Head are significant, and should be studied further in order to improve our understanding of the $\Delta'(^{17}O)$ budget in atmospheric $CO_2$. We think the records, together with the comparison of the measurements and the model simulation we present in the manuscript contain valuable information on the potential influence of the stratospheric input of $\Delta'(^{17}O)$ on the total budget of $\Delta'(^{17}O)$ of $CO_2$ in the troposphere. Below we react on the referee's comments in the bold and italic text.

Major comments:

1. Need to have a paragraph summarizing the errors/biases of SICAS and possible sampling/storage biases. The SICAS D17O measurements/results are suspicious. Detail analysis of IRMS D17O, though limited, is not available. See specific comments below.

   **We understand that the reviewer is skeptical, but we do not agree with the qualification that our results are suspicious, and that limited details of this technique are available. To help the discussion on this forward, we reiterate some of it here and also added extra text to the revised manuscript to help the reader appreciate the technique, and its uncertainties, better.**

   **For more elaborate technical details on the SICAS errors/biases and possible sampling/storage biases we refer to Steur et al. (2021), Steur et al. (2023) and Steur (2023), but similar measurement and calibration techniques can also be found in Bajnai et al. (2023), Hare et al. (2022) and Perdue et al. (2022). In the manuscript we only discuss these matters briefly, as our main aim is to present the measurement records, not to describe the technical details of the measurement process.**

   **The main reason for higher uncertainties of the SICAS measurements is due to $CO_2$ amount fraction dependencies, also identified in Bajnai et al. (2023), which are hard to correct when the samples are outside the range of our reference cylinders. We added the following text in line 140 that elaborates on this:**

*"We calibrate with the reference cylinders only, instead of having an on-line mixing facility where the reference and sample $CO_2$ amount fraction can be matched (Perdue et al., 2022; Bajnai et al., 2023). Therefore, samples that fall outside the range of the $CO_2$ amount fraction that is covered by the reference cylinders will have higher uncertainties."*

*We thereby added the following outlook on how to possibly deal with this in the future (line 171): "Extending the $CO_2$ amount fraction range of our reference cylinders will improve the measurement precision of samples with elevated $CO_2$ amount fractions, as well as extend the range of $CO_2$ amount fractions that can be shown in the results. A way to prevent that a high number of reference cylinders has to be included at all times, is to make the selection of references more dynamic. As sample measurements are always alternated with a working gas measurement, it is possible to do a 1-point calibration immediately after a sample is measured. In this way it will be possible to select the ideal set of references to calibrate the samples based on the $CO_2$ amount fractions derived from the 1-point calibration. This would save reference gas, as well as measurement time of a measurement series."*

*The combined uncertainty includes the introduced uncertainty as result of the calibration process, based on analysis of reference gases over a broad range of $CO_2$ amount fractions over a period of 2 years (Steur, 2023). Sampling biases on oxygen istotopes are studied in Steur et al. (2023), which we refer to in the manuscript. Sampling biases on the $\Delta'(^{17}O)$ specifically are discussed in the manuscript in line 227 where we state:*

*"These high differences are connected to the observations of drift in the oxygen isotopes of $CO_2$ in flask samples as a function of time (Steur et al., 2023). $\Delta'(^{17}O)$ values are not (or hardly) affected by the drifts in oxygen isotopes in the flasks. We calculated that, in the extreme case of a change of more than 3 ‰ in $\delta(^{18}O)$ of atmospheric $CO_2$ (Steur, 2023) resulting from equilibration of $CO_2$ with water inside the flask, and at the same time an initial $\Delta'(^{17}O)$ value of the $CO_2$ of -0.69 ‰, changes the $\Delta'(^{17}O)$ less than 0.06 ‰. Considering that the uncertainty of the SICAS $\Delta'(^{17}O)$ measurements is always 0.05 ‰ or higher, we can conclude that the effect of drift of the oxygen isotopes inside the flasks is negligible for the $\Delta'(^{17}O)$ values. Results and calculations that support this conclusion can be found in Appendix B1"*

2. Keeling binary-mixing analysis (and Keeling plots) is suggested to be made, to understand the endmembers, if any, controlling the variations of the isotope data. Color-coded diagram is hard to see. Scatter plots of D17O vs. d13C and D17O vs. conc(CO2) can be used to understand how much the variation in D17O is due to anthropogenic (e.g., see Liang et al., AAQR, 2017). Anthropogenic contribution (or even stratospheric influence) can also be assessed by comparing CO2 (including its isotopologues) and CO. This exercise is essential to tell whether the CO2 isotope data contain useful information, or just noise/errors from the measurements.

*Based also on a request from reviewer #1, the Lutjewad $\Delta'(^{17}O)$ summer and winter values plotted against $\delta(^{13}C)$ and $1/CO_2$ are added in Appendix D. These plots were used to show that no seasonal cycle can be detected from the Lutjewad $\Delta'(^{17}O)$ record. We would expect fossil fuel emissions to appear in the winter values as negative $\Delta'(^{17}O)$ values should correlate in this case with high $CO_2$ and low $\delta(^{13}C)$ values. However, in our dataset high $CO_2$ values are not shown due to the range in $CO_2$ amount fraction of our reference cylinders. We changed the text in line 350 accordingly:*

*"The most important difference between the Lutjewad and Mace Head $\Delta'(^{17}O)$ records is the presence of more depleted values in the Lutjewad record, with the lowest value being -0.43 ‰ in the summer of 2022. $CO_2$ equilibrated with water following $\lambda_{RL}$ will have an $\Delta'(^{17}O)$ of -0.21 ‰. In summer, leaf water gets enriched in oxygen isotopes, and depleted in $\Delta'(^{17}O)$ as the result of high rates of evapotranspiration (Landais et al., 2006). Due to the active biosphere during summer, $CO_2$ and leaf water will equilibrate and the depleted $\Delta'(^{17}O)$ signal will be translated to the $CO_2$ (Adnew et al., 2023). We estimated that this could result in $\Delta'(^{17}O)$ values being up to 0.1 ‰ more depleted, when assuming the minimum $\theta$ of 0.516 for evapotranspiration (Landais et al., 2006), and considering the range of $\delta(^{18}O)$ values that were measured in our Lutjewad record. For the full estimation we refer to Appendix E. $\Delta'(^{17}O)$ values up to -0.31 ‰ can be explained by this process. $CO_2$ emitted from combustion processes has very negative $\Delta'(^{17}O)$ values (Laskar et al., 2016; Horváth et al., 2012). All points that have lower $\Delta'(^{17}O)$ than -0.3 ‰, and are sampled during winter/spring, have more depleted $\delta(^{13}C)$ values and more enriched $CO_2$ values than would be expected from the seasonal trends. This indicates that local $CO_2$ emission sources are the reason for the more depleted $\Delta'(^{17}O)$ values in winter. Samples that are very enriched in $CO_2$ amount fractions are not shown here, as that results in very high measurement uncertainties. This could be the reason that a correlation of $\Delta'(^{17}O)$ and $CO_2$ amount fractions does not appear in figure D1. A few points show depletions lower than -0.31 ‰ without $CO_2$ amount fraction enrichments, and remain for now unexplained."*

3. Need a more detail discussion on the modeling. Are the changes mainly in the D17O value in the downwelling flux or the changes are due mainly to the enhanced flux with D17O value little changed? For Eq(11), please elaborate it further. How much contribution is from the newly added 100 mbar temperature term? Is the term the anomaly from the climatology temperature? Please define "anomaly." Please compare with PV and/O3 at 100 mbar. What is the source of 0.08 per mil mentioned in Fig 7 caption? If it's from the newly added term, does it mean that the D17O from the model stratosphere is biased too high?

*In the adjusted model we add the $\Delta T$ term which will change the $\Delta'(^{17}O)$ value in the stratosphere It will, however, also have an effect on the*

*stratosphere-troposphere exchange. We argue in the text (line 416): "Note that the temperature relation represents both temperature dependence of the actual Δ'($^{17}$O) as suggested in Wiegel et al. (2013) and the temperature dependence in stratospheric exchange, which might not be sufficiently represented with only 25 vertical layers in the current model (see e.g. Bânda et al., 2015, for the influence of vertical resolution on stratosphere-troposphere exchange)." It is easier to change the $\Delta'(^{17}O)$ production term than to adjust the air mass, as this will lead to inconsistencies in the model. The model simulation as shown in this study is therefore (still) not a correct representation of the stratospheric $\Delta^{17}O$ budget, but does show the need to include larger variability of this term in the model for a correct representation.*

*The difference between the original model simulation and the adjusted model simulation can thus be fully attributed to the added 100 hPa term.*

*The definition of "anomaly" should indeed be given and is now added to the text in line 413:*

*"The temperature anomaly $\Delta T_{100hPa}$ is determined by taking the average temperature of the months January, February and March at 100 hPa for 60-90° N per year for the period 2017-2022. Subsequently the difference between these values and the average of all 6 years is calculated."*

4. Figure 7: mid-year peak in most of the years except 2020, due to enhanced STE in spring, mentioned in the text. What is the cause of the missing peak in this particular year? Also what is source mechanism causing D17O less than 0? If I understand correctly, one has to subtract 0.08 per mil from the modified model, inconsistent with the statement -0.061-0.056 per mil variation range mentioned in Line 400. Does this mean the model was not appropriately made?

*There are two source mechanisms for an $\Delta'(_{17}O)$ of less than -0.21 ‰, as we now use a $\lambda$ of 0.528 for expression of $\Delta'(_{17}O)$, are discussed in the text. When depleted $\Delta'(_{17}O)$ values also show enriched $CO_2$ amount fractions compared to the background curve and depleted $d^{13}C$ values it is very likely to be due to fossil fuel emissions. When depleted $\Delta'(_{17}O)$ values show lower $CO_2$ amount fractions in comparison with the background curve, high rates evapotranspiration can explain the depleted $\Delta'(_{17}O)$ values. We argue that high rates of evapotranspiration can account for the values of $\Delta'(_{17}O)$ up to -0.31 ‰, when there is an active biosphere at that time. This is in the text line 350:*

*"The most important difference between the Lutjewad and Mace Head $\Delta'(_{17}O)$ records is the presence of more depleted values in the Lutjewad record, with the lowest value being -0.43 ‰ in the summer of 2022. $CO_2$*

*equilibrated with water following λ$_{RL}$ will have an Δ'($_{17}$O) of -0.21 ‰. In summer, leaf water gets enriched in oxygen isotopes, and depleted in Δ'($_{17}$O) as the result of high rates of evapotranspiration (Landais et al., 2006). Due to the active biosphere during summer, CO$_2$ and leaf water will equilibrate and the depleted Δ'($_{17}$O) signal will be translated to the CO$_2$ (Adnew et al., 2023). This could result in Δ'($_{17}$O) values being up to 0.1 ‰ more depleted, when assuming the minimum θ of 0.516 for evapotranspiration (Landais et al., 2006), and considering the range of δ($_{18}$O) values that were measured in our Lutjewad record. Δ'($_{17}$O) values up to -0.31 ‰ can therefore be explained by this process. CO$_2$ emitted from combustion processes has very negative Δ'($_{17}$O) values (Laskar et al., 2016; Horváth et al., 2012). All points that have lower Δ'($_{17}$O) than -0.3 ‰, and are sampled during winter/spring, have more depleted δ($_{13}$C) values and more enriched CO$_2$ values than would be expected from the seasonal trends. This indicates that local CO$_2$ emission sources are the reason for the more depleted Δ'($_{17}$O) values in winter. Samples that are very enriched in CO$_2$ amount fractions are not shown here, as that results in very high measurement uncertainties. This could be the reason that a correlation of Δ'($_{17}$O) and CO$_2$ amount fractions does not appear in figure D1. The source mechanism for points not showing a CO$_2$ amount fraction enrichment that are more depleted than -0.31 ‰ remains unexplained."*

*The numbers that are given in the text are adjusted to the actual outcomes of the model simulation, without subtraction of 0.08 ‰, as well as these numbers are now shown in figure 7 to avoid confusion. We focus on the total variability and the timing of the peaks/throughs, as "the long-term mean values simulated by the model for Lutjewad are ultimately dependent on the integrated contribution from all processes across the globe, which are poorly constrained in the model (e.g. due to large uncertainties in soil exchange, see Wingate et al. (2009))." (line 247).*

*The adjusted text giving the total range of the measurements and the model simulation are now in line 419: "The overall variability over the full record is -0.19 to -0.07 ‰ for the model simulation and -0.27 to -0.16 ‰ for the moving average of the measurements. Although the absolute values of the measurements and the model differ by 0.08 ‰, the overall variability of the simulation with the adjusted Δ'($_{17}$O) production term increased significantly and is close to the overall variability of the measurements."*

Other comments:

1. The CO2-O2 exchange method for D17O measurements was first developed by Mahata et al., not Adnew et al. Please acknowledge the previous effort.

   *The reference is added.*

2. make needed correction/clarification to small delta and big Delta in the presentation in the Introduction section.

*We revised the use of the small and big delta in the whole manuscript as, indeed, the capital delta was sometimes used incorrectly, as we should have used a small delta.*

3. Line 54: rephrase/elaborate 10 per meg for reference gas measurements. Do you mean 10 ppm is achieved for "reference" gas only?

*The sentence is rephrased and we added another reference where the same method was used and similar precisions are reached: "The last method mentioned is at this moment acquiring a precision being better than 10 per meg for measurements of $\Delta'(_{17}O)$ (Adnew et al., 2019; Liang et al., 2023)." (line 54)*

4. Line 70: Please include Liang et al. (2023, Scientific Reports) who reported an updated data set that also include new data from Palos Verdes peninsula, CA.

*The reference was added.*

5. Line 105-106: Rephrase/elaborate "the stability of trace gas amount fractions." It is not clear whether you referred to CRDS instrumental precision/stability or the concentrations of the gases of interest in the atmosphere/flasks.

*Rephrased as "CRDS continuous measurements are shown as hourly means and therefore the standard deviations can vary considerably, depending on the stability of trace gas amount fractions in the atmosphere during the measurement period". (line 108)*

6. Trace gas concentration and isotope measurements: are the measurements made for the same flasks collected?

*Rephrased the first sentence of section 2.3 (line 115) to: "Stable isotope composition measurements are conducted directly on atmospheric air samples, on the same flasks collected for the trace gas amount fraction measurements,"*

7. Line 181-186: Are the D(17O) D(17O) or d(17O)?

*These should indeed be $\delta(^{17}O)$, this was changed in the text.*

8. Section 2.4, first paragraph. I believed you meant to compare SICAS with DI-IRMS. The first sentence seemed to say that you compared SICAS at CIO with that at IMAU. Please rephrase and make needed correction/clarification.

*The sentence was rephrased to (line 214): " For a selection of Lutjewad samples two flasks containing identical air were sampled (from now defined as a duplo) of which one flask was measured at the CIO using laser*

*absorption spectroscopy and one at the IMAU using DI-IRMS to check the compatibility of the two methods."*

9. Section 2.4. Figure 2 caption: how is the "combined" uncertainty defined and source of errors? Is the length of the error bar 1-sigma or +/- 1-sigma? Please define it clearly. Are the errors in the difference mainly from CIO? Why the extraction at IMAU is more variable?

   *The combined uncertainty for the SICAS measurements is the same uncertainty as defined in section 2.3 (now added to the caption) and the length of the error bars is +/- 1-sigma (added to all captions). The measurement uncertainty of the IMAU measurements is considerably lower (about 0.01 ‰) and is therefore not shown. For the $\Delta^{17}O$ measurements, the majority of the points have no difference, when considering the uncertainty budget of the CIO. All points fall within the borders of ±0.05 ‰ difference, taking again into account the uncertainty budget of the CIO. For the $\delta^{13}C$ values the differences are higher and there are multiple points that fall outside the ±0.03 ‰ difference. It is hard to say what the exact cause for the differences is, as several processes can cause differences in the results from both labs: sampling procedures, storage procedures, $CO_2$ extraction procedures, as well as the different measurement methods.*

10. Figure 7: Is 0.08 per mil from the model? What's the source/cause of this?

    *We focus on the total variability and the timing of the peaks/throughs, as "the long-term mean values simulated by the model for Lutjewad are ultimately dependent on the integrated contribution from all processes across the globe, which are poorly constrained in the model (e.g. due to large uncertainties in soil exchange, see Wingate et al. (2009))." (line 252).*

11. Section 3.1 last paragraph. From Figure 3, I don't see "clearly" the drought points mentioned. Normally I'd expect drought would reduce biospheric uptake and thus cause CO2 increase. Here it said the opposite that the decrease in May-June 2018 was due to the drought. I would suggest to have a separate figure showing the deviation from the average (the background) and discuss the cause of the deviation, such as droughts, in more detail. Also there are two NOAA points next to the referred Lut(CO2) reduction, and that can be used to support the reduction.

    *We rephrased this paragraph, as the text was indeed not very clear. Also, the two NOAA points are now mentioned in the text to support the statement. Line 283: "The Europe wide drought, which was most severe in Northern Europe, during the summer of 2018 (Peters et al., 2020; Ramonet et al., 2020) is clearly visible in the continuous $CO_2$ amount fraction record of Lutjewad, where a short-term increase in $CO_2$ amount fractions interrupts the overall decrease in amount fractions that normally occurs over the growing season. In early spring of 2018, $CO_2$ amount fractions decrease*

*rapidly (when the growing conditions were more favorable, see Smith et al., (2020)), until May 2018. Subsequently a rapid increase in $CO_2$ amount fractions is observed that lasts until June, before $CO_2$ amount fractions start decreasing again. This event is only visible in one Lutjewad flask sample having a $\Delta_{bg}y(CO_2)$ of -8.6 µmol/mol and two Mace Head samples from the NOAA-GML CCGG having $\Delta_{bg}y(CO_2)$'s of -6.7 and -7.1 µmol/mol."*

12. Line 328: D17O is affected little by transpiration. It's mainly due to evaporation, or evapotranspiration.

   *Changed transpiration to evapotranspiration.*

13. Line 395: better agreement "than" the ...

   *Changed to "than".*

14. Line 473: I believe here you meant d17O, not D17O.

   *We indeed meant here $\delta(^{17}O)$, which is now changed. Also we added the $\Delta'(^{17}O)$ value of the water. Line 498: "For the initial $\delta(_{18}O)$ and $\delta(_{17}O)$ of the water we use -12.91 and -6.77 ‰ VSMOW, respectively. The $\Delta'(_{17}O)$ value of the water is 0.07 ‰."*

15. Appendix B: Are the results experimental results or from model simulation? If they are experimental, please provide measurement errors? What's the D17O value of the water? With that, is the change in D17O reflected in d18O? That is, is the co-variation of d18O and D17O following water-CO2 equilibration line?

   *The results are from a model simulation, as stated in the first sentence in Appendix B (line 490): "To determine the change in $\Delta'(_{17}O)$ as the result of drift of the oxygen isotopes of atmospheric $CO_2$ inside glass sample flasks (Steur et al., 2023), a simulation of the various changes was conducted."*

   *The $\Delta'(^{17}O)$ values were added in the text in line 498 (see the answer above).*

16. Figure C1 and App C: Other than the two lowest SICAS points, there is no correlation between SICAS D17O and IRMS D17O. IRMS higher precision measurements show a factor of ~3 more variation than SICAS. IRMS as claimed has higher precision. One has to discuss whether the large variation in D17O is also seen in and supported by other data, such as CO2 (conc, d13, d18O) and CO.

   *We need to consider that we are looking at samples that were measured under different sample preparation methods. Part of the flasks that were measured by DI-IRMS at IMAU was extracted at the CIO, part of the flasks was extracted at the IMAU. Besides that, the low variance in the $\Delta'(^{17}O)$ in*

*the set of duplicate flasks, not more than 0.15 ‰, and the average combined uncertainty of the SICAS measurements of 0.07 ‰, also complicates the comparison between IMAU and CIO measurements. Figure C2 does, however, show that the general trend in $\Delta'(^{17}O)$ values measured at Lutjewad is reflected by measurements from both labs.*

---

## Author Response (AR2)

**Answer to editor's report**

Dear Jan Kaiser,

We thank you for the time and effort that was put in editing the manuscript. The comments raised by the reviewers are addressed in separate reports. The corrections that were proposed by you are addressed below, with the answer in **bold and italic**:

1) A lowercase theta is generally preferred over an uppercase one for the triple isotope fractionatio coefficient (Eq. 2).

***The lowercase theta is used now throughout the manuscript.***

2) You are free to use $\Delta(17O)$ symbol without a "prime" since there is no distinction necessary.

***Thank you for this suggestion. We now use the symbol without a prime throughout the manuscript.***

3) L. 129: $\Delta$ should be italics.

***The symbol is changed to italics.***

4) The amount fraction symbol y should be in italics.

***The amount fraction symbol y is now in italics throughout the text.***

5) Table 1, heading column 1 should be $y(CO2)$ / ($\mu$mol mol$^{-1}$). Column 2 should be $\delta$(13C, VPDB) / ‰ [or $\delta$_VPB(13C) / ‰, as in Eq. 5]. Similarly, for $\delta$(17O), $\delta$(18O).

***The column headings are changed as suggested.***

6) Eq. 4: The symbols on the left side of the equal sign should be $u(\delta(13C))$/‰. On the right side, you should write $\Delta y(CO2)$/($\mu$mol mol$^{-1}$) and omit the symbol ‰ from the last term, to make the equations dimensionally correct.

***We thank the editor for these corrections, which are now in the manuscript.***

7) Figure A2 caption: spelling of triangles

***The spelling is corrected.***

Answer to referee #1:

We thank you very much for the comments and suggestions that improved the manuscript earlier, as well as the positive referee report after these revisions. The last comments of the referee are addresses in this report, with our answers in **bold and italic**.

In the conclusion, the authors state that:
"Our results show that the biosphere is not the dominant process for variations in Δ'(17O). Δ'(17O) of atmospheric CO2 is therefore not suitable as a proxy for quantification for gross primary production at our study locations. The variation in the stratospheric source of high Δ'(17O) is possibly the cause for the high interannual variations we observe in the records." The data and model presented by the authors highlight the significance of the stratospheric signal. However, it's possible that more precise and higher-resolution data could reveal a bisphere signal. Therefore, I suggest that the authors focus on what can be resolved instead of dwelling on what cannot, and remove the sentence from their conclusions regarding the unsuitability of Δ'17O measurements for GPP estimations.

***We thank the referee for this suggestion. We deleted the last sentences of our conclusion, that indeed, would fit better in the discussion.***

In Line 207, the authors write: "Note that the scale described above for the Δ'(17O) values is indirectly linked to VSMOW, adding uncertainty to the compatibility of other Δ'(17O) scales." No internationally accepted reference material is yet available for Δ'17O in CO2. This means that the CO2 measurements are not directly linked to the VSMOW water reference, and there are many steps involved that can lead to uncertainty. For example, one could anchor their data to measurements of O2 gas from either fluorinating a carbonate reference material or to CO2 equilibrated with VSMOW. It would be useful if the authors could provide more information on the steps taken to link their values to the VSMOW scale, as this would help make the dataset more future-proof.

***We realize that the scale that is used for our $\Delta^{17}O$ (and $\delta^{17}O$) values is not ideal. Measurement values from the BGC-IsoLab in Jena were used in combination with measurement values from the IMAU, described in lines 195-208. We chose for this approach as we are dealing with CO2 in air samples, and wish to keep the connection to the JRAS scale, which is realized by the BGC-IsoLab in Jena. The steps we take for connecting our measurements results to the VSMOW scale are, we think, described clearly in this manuscript in lines 192 to 212. The calibration method from the IMAU measurements can be found in Adnew et al. (2019), which is given in the manuscript.***

In the legend of Figure 3, I suggest specifying the location of the "Continuous monitoring".

***We specified the location of the continuous monitoring points, and added "Flasks" to all other data categories to clarify the figure.***

In Figure 4, there is an inconsistency in the line widths.

**The inconsistency is removed in figure 4.**

We want to thank the referee for his/her time and effort he/her put in commenting on the first manuscript, which did help to improve the manuscript, especially in more elaborately discussing the limitations of the dataset, as well as give an outlook for improving the measurements in the future. We are happy with the final report of the referee, and we answer the final comments and suggestions that were given in this report. Our answers are in **bold and italic:**

High D17O variability could be explained by enhanced D17O values from the stratosphere. I agree that due to the precision achieved at the moment, the difference/variation of ~0.1-0.15 per mil cannot be resolved easily. I also agree that Figure C2 is a better way to show/demonstrate the spectroscopy data can be made useful and show that SICAS D17O pattern is very likely to be real. Please make this clearer, for example, by emphasizing this in the main text (not just in the appendix) and also the caption of Fig C1.

***Thank you very much for these suggestions. We agree that figure C2 is convincing in the observed pattern of the SICAS $\Delta^{17}O$ measurements. In the main text, this is stated in lines 240-242. We added the specific reference to figure C2 in line 241 to make it clearer, as suggested.***

***Then we state the following in appendix C (lines 511-513): "When considering, however, all datapoints of the Lutjewad flasks from the SICAS and from the IMAU, significant interannual variability is reflected in both datasets. Both the IMAU and the SICAS measurements show lower values in 2020 than in the period before, as observed in figure C2."***

***We therefore did not add this information in the caption of figure C1, to avoid repetition.***

0.08 per mil elevation in the model with dT term included: How much contribution is the newly added term compared to the previous one from N2O? I did a quick estimate from, for example, Koren et al. (2019, JGR) Figure 1, the newly added term is comparable to the existing term, obtained by taking D17O=0.7 per mil from the stratosphere and a factor of ~10 dilution from the land and ocean and the 0.08 per mil increase. That is to say that to explain the data with the newly added term, one needs a lot more CO2 from the ocean (as the site is near the coast and the authors also claimed from absence of correlation between, for example, d13C and conc in CO2; this statement is confusing, see more below), especially to explain the changes from mid-2019 to late 2020, to bring the model values down to match the data. An alternative is to reduce the contribution from the first term, the N2O (eq. 8), which also does not show significant interannual changes. But to do that may change the correlation of CO2 D17O and N2O in the stratosphere. This has to be checked and discussed, at least briefly if not possible to be made thoroughly.

***We thank the referee for these comments and suggestions. We agree that there is an offset between the revised model run and the measured values. However, we think the revised model run should be considered as a very first, conceptual model to show that stratospheric input of $\Delta^{17}O$ is more variable than previously described. The mechanism***

*that is now used in both the original and revised model, is to impose empirical relations for the value of $\Delta^{17}O$ in the simulated stratosphere following simulated $N_2O$ and temperature. In reality, there will be a combination of change of both the $\Delta^{17}O$ value and the stratosphere-troposphere transport. We now state this more clearly in the text, line 416: "Note that both the $N_2O$ and the temperature anomaly are used as proxy values for the $\Delta(^{17}O)$ in the stratosphere. Thereby, the temperature relation represents both temperature dependence of the actual $\Delta(^{17}O)$ as suggested in Wiegel et al. (2013) and the temperature dependence in stratospheric exchange, which might not be sufficiently represented with only 25 vertical layers in the current model (see e.g. Bândă et al., 2015, for the influence of vertical resolution on stratosphere-troposphere exchange)."*

Biological uptake

Line 327-328: It says "We did not observe any of this, indicating there is no significant biosphere signal in our Lutjewad Δ'(17O) record." This is inconsistent with that described in the paragraph starting at Line 350, that evapotranspiration is taken to explain the reduced D17O during summer time; usually high evapotranspiration is associated with high biological uptake/cycling. How's wind direction during CO2 sampling time? Does it support terrestrial origins of CO2 (which could be affected more easily by evapotranspiration processes)?

*Thank you for this comment. We agree that the statement is inconsistent, and we should formulate it differently. It is now formulated as follows:*

*Line 327: "We did not observe any of this, indicating there is no significant seasonality caused by the biosphere signal in our Lutjewad Δ'(17O) record."*

Also super low values less than -0.3 per mil, due to anthropogenic? d13C and [CO2] are better measured. How much reduction is needed in d13C and [CO2], if it's from combustion? Is it supported by CO?

*The answer to this question is given below.*

For better understanding the CO2 isotope data and variability, please show Keeling analysis results for d13C and [CO2] and briefly discuss the endmember obtained. As claimed in the text, very low values of D17O are likely due to combustion; please highlight the data points having low D17O values.

*$CO_2$ and d13C values are discussed in the text in relation to the low D17O values, please see line 359: "All points that have lower Δ(17O) than -0.3 ‰, and are sampled during winter/spring, have more depleted δ(13C) values and more enriched CO2 values than would be expected from the seasonal trends. This indicates that local CO2 emission sources are the reason for the more depleted Δ(17O) values in winter. Samples that are very enriched in CO2 amount fractions are not shown here, as these results have very high measurement uncertainties. This could be the reason that a correlation of Δ(17O) and CO2 amount fractions does not appear in figure D1."*

*We don't want to overanalyse the data, as we know there is a bias towards the high CO2 values. SICAS measurements with high CO2 values (far outside the range that is covered by the calibration cylinders) are not included in the results due to their high uncertainties. We therefore chose to not make any Keeling plots, as our dataset will be a limited representation of the different endmembers.*

Line 73: peninsula

*Line 73 was corrected as suggested.*

Line 322: Suggest to replace Hoag et al. by Koren et al.; Hoag et al. did not study seasonal cycle explicitly, even though it can be inferred.

*We changed the reference as suggested.*